# Change from aerosol-driven to cloud feedback-driven trend in shortwave radiative flux over the North Atlantic

Daniel P. Grosvenor[1] and Kenneth S. Carslaw[2]

[1]National Centre for Atmospheric Sciences, School of Earth and Environment, University of Leeds, Leeds, LS2 9JT, UK
[2]Institute for Climate and Atmospheric Science, School of Earth and Environment, University of Leeds, Leeds, LS2 9JT, UK

**Correspondence:** D. P. Grosvenor
(d.grosvenor@leeds.ac.uk)

**Abstract.**

Aerosol radiative forcing and cloud-climate feedbacks each have a large effect on climate, mainly through modification of solar shortwave radiative fluxes. Here we determine what causes the long-term trends in the upwelling shortwave (SW) top-of-the-atmosphere (TOA) fluxes ($F_{SW\uparrow}$) over the North Atlantic region. Coupled atmosphere-ocean simulations from the UK Earth System Model (UKESM1) and the Hadley Centre General Environment Model (HadGEM3-GC3.1) show a positive $F_{SW\uparrow}$ trend between 1850 and 1970 (increasing SW reflection) and a negative trend between 1970 and 2014. We find that the 1850-1970 positive $F_{SW\uparrow}$ trend is mainly driven by an increase in cloud droplet number concentration due to increases in aerosol, while the 1970-2014 trend is mainly driven by a decrease in cloud fraction, which we attribute mainly to cloud feedbacks caused by greenhouse gas-induced warming. In the 1850-1970 period, aerosol-induced cooling and greenhouse gas warming roughly counteract each other so that the temperature-driven cloud feedback effect on the $F_{SW\uparrow}$ trend is weak (contributing to only 23% of the $\Delta F_{SW\uparrow}$) and aerosol forcing is the dominant effect (77% of $\Delta F_{SW\uparrow}$). However, in the 1970-2014 period the warming from greenhouse gases intensifies and the cooling from aerosol radiative forcing reduces, resulting in a large overall warming and a reduction in $F_{SW\uparrow}$ that is mainly driven by cloud feedbacks (87% of $\Delta F_{SW\uparrow}$). The results suggest that it is difficult to use satellite observations in the post-1970 period to evaluate and constrain the magnitude of the aerosol-cloud interaction forcing, but that cloud feedbacks might be evaluated.

Comparisons with observations between 1985 and 2014 show that the simulated reduction in $F_{SW\uparrow}$ and the increase in temperature are too strong. However, the temperature discrepancy can account for only part of the $F_{SW\uparrow}$ discrepancy given the estimated model feedback strength ($\lambda = \frac{\partial F_{SW}}{\partial T}$). The remaining discrepancy suggests a model bias in either $\lambda$ or in the strength of the aerosol forcing (aerosols are reducing during this time period) to explain the too-strong decrease in $F_{SW\uparrow}$, with a $\lambda$ bias being the most likely. Both of these biases would also tend to cause too-large an increase in temperature over the 1985-2014 period, which would be consistent with the sign of the model temperature bias reported here. Either of these model biases would have important implications for future climate projections using these models.

# 1 Introduction

Many changes have occurred over the historical period, 1850–2014: the industrial revolution, during which North America and Europe in particular emitted increasing amounts of greenhouse gases, aerosols and aerosol precursors; the introduction of clean air acts by North America and Europe starting in the 1950s that led to reduced aerosol emissions from those regions; the industrialisation of China and India leading to increased emissions of greenhouse gases and aerosols; and the general continued rise in the rate of global greenhouse gas emissions. Climate change over the North Atlantic (NA) on decadal and longer timescales is influenced by many different factors, with the most significant likely being greenhouse gas forcing, aerosol forcing, mid-latitude cloud feedbacks mediated by temperature changes, natural variability and the Atlantic meridional overturning circulation (AMOC). Many of the processes involve changes in the upwelling shortwave (SW) radiative flux at the top of the atmosphere ($F_{SW\uparrow}$), therefore $F_{SW\uparrow}$ is a key property of the Earth system when considering climate variability.

Fairly long-term observational records of $F_{SW\uparrow}$ exist for the recent part of the historical period (e.g., 1985-2019; Allan et al., 2014a; Liu et al., 2015, 2017) that may be useful for evaluating models and attributing changes in climate to various causes. However, to understand what model evaluation using such long-term datasets is telling us about the causes of regional climate change it is necessary to understand the driving factors of long-term changes in $F_{SW\uparrow}$. In this study we use the UK Met Office climate models to better understand the underlying processes and what the observed long-term trends in $F_{SW\uparrow}$ might be telling us about model performance and causes of regional climate change.

We focus on the NA region because it plays a major role in several aspects of the Earth system. The NA ocean sequesters large amounts of carbon and heat from the atmosphere and therefore helps to regulate the global climate (Buckley and Marshall, 2016). Processes in the NA are thought to help determine the speed of the Atlantic meridional overturning circulation (AMOC; Buckley and Marshall, 2016), which transports a significant amount of heat northwards representing ∼25% of the total (atmospheric plus oceanic) global northward heat transport at 24-26º N (Srokosz et al., 2012). The AMOC transports a large amount of energy from the southern hemisphere to the northern hemisphere, something that is not true for the equivalent circulations in the Pacific (Buckley and Marshall, 2016). This cross-equatorial oceanic heat flow is important because it leads to a compensating southward cross-equator heat flow within the atmosphere and this in turn causes the Intertropical Convergence Zone (ITCZ) to be positioned north of the equator. Changes in the AMOC can therefore lead to changes in the ITCZ position, which could bring great disruption to the climate of not just the Atlantic region, but also the climates of the global tropics, sub-tropics, and potentially the mid-latitudes via changes in precipitation and changes to the Indian and Asian monsoons (Buckley and Marshall, 2016; Chiang and Friedman, 2012; Srokosz et al., 2012).

The NA is surrounded by North and Central America, Europe and North Africa which are large regions of high population density. This means that (1) there is a great deal of influence from short-lived anthropogenic species such as aerosols over the NA, and (2) that changes in the NA climate system can have large impacts on human society. Sea surface temperature (SST) variability in the NA has been associated with impacts on important phenomena such as tropical storm and hurricane activity (Zhang and Delworth, 2006; Smith et al., 2010; Dunstone et al., 2013); anomalies in rainfall in Europe and North America (Sutton and Hodson, 2005; Sutton and Dong, 2012); African Sahel and Amazonian droughts (Hoerling et al., 2006; Knight

et al., 2006; Ackerley et al., 2011); Greenland ice-sheet melt rates (Holland et al., 2008; Hanna et al., 2012); sea-level anomalies (McCarthy et al., 2015); and the strength of the mid-latitude jet (Woollings et al., 2015). Robson et al. (2018) provides a review of changes in the North Atlantic climate system, with a focus on more recent changes.

Aerosol effective radiative forcing ($\Delta F_{\mathrm{aer}}^{\mathrm{eff}}$) is a key driver of long-term changes in $F_{\mathrm{SW\uparrow}}$ over the NA and globally. For the calculation of $\Delta F_{\mathrm{aer}}^{\mathrm{eff}}$ all physical variables are allowed to respond to perturbations except for those concerning the ocean and sea ice, (e.g., see Myhre et al., 2013), meaning that surface temperatures need to be constant. As discussed further below, this makes $\Delta F_{\mathrm{aer}}^{\mathrm{eff}}$ difficult to calculate using timeseries from observations or coupled climate models since radiative fluxes respond to changes in temperature, for example, due to cloud feedbacks. $F_{\mathrm{SW\uparrow}}$ is a focus for aerosol forcing because aerosol forcing

generally occurs through the effect of aerosols on shortwave radiative fluxes rather than longwave fluxes; e.g., O'Connor et al. (2021) estimates a global SW $\Delta F_{\mathrm{aer}}^{\mathrm{eff}}$ of -1.26 W m$^{-2}$ and a longwave $\Delta F_{\mathrm{aer}}^{\mathrm{eff}}$ of 0.17 W m$^{-2}$ for the UKESM1 (UK Earth System Model v1). Henceforth in this paper we only consider shortwave fluxes, forcings and feedbacks.

   $\Delta F_{\mathrm{aer}}^{\mathrm{eff}}$ can be separated into a component due to aerosol radiation interactions (ARI) that occurs in cloud-free air ($\Delta F_{\mathrm{ari}}^{\mathrm{eff}}$; sometimes also known as the direct effect) and a component due to aerosol-cloud interactions (ACI, or indirect effects) des-

ignated as $\Delta F_{\mathrm{aci}}^{\mathrm{eff}}$. The ACI component of $\Delta F_{\mathrm{aer}}^{\mathrm{eff}}$ can also be broken down into two further components. First, that due to an change in cloud droplet concentration ($N_{\mathrm{d}}$) at constant liquid water content (LWC) and constant cloud fraction ($f_{\mathrm{c}}$), which causes a change in the cloud droplet effective radius ($r_{\mathrm{e}}$) and hence cloud albedo. Here we will designate this component $\Delta F_{\mathrm{aci\ N_d}}^{\mathrm{eff}}$, often termed the instantaneous radiative forcing, or the Twomey effect (Twomey, 1977). Second, rapid adjustments of LWC (or the vertical integral of this, which is the liquid water path, $L$), and/or adjustments in $f_{\mathrm{c}}$ that occur in response to

the initial decrease in droplet size associated with the $N_{\mathrm{d}}$ increase. Note here that we define $L$ to be the in-cloud value, not the mean of a partly cloudy sky. We designate the forcings from these adjustments as $\Delta F_{\mathrm{aci\ L}}^{\mathrm{eff}}$ and $\Delta F_{\mathrm{aci\ f_c}}^{\mathrm{eff}}$ and note that $\Delta F_{\mathrm{aci}}^{\mathrm{eff}} \approx \Delta F_{\mathrm{aci\ N_d}}^{\mathrm{eff}} + \Delta F_{\mathrm{aci\ L}}^{\mathrm{eff}} + \Delta F_{\mathrm{aci\ f_c}}^{\mathrm{eff}}$. The mechanisms that cause these adjustments involve several microphysical and thermodynamical processes (Albrecht, 1989; Stevens et al., 1998; Ackerman et al., 2004; Bretherton et al., 2007; Hill et al., 2009; Berner et al., 2013; Feingold et al., 2015). For regions of the NA north of 18º N UKESM1 suggests that $\Delta F_{\mathrm{aci}}^{\mathrm{eff}}$ greatly dominates over

$\Delta F_{\mathrm{ari}}^{\mathrm{eff}}$ (Grosvenor and Carslaw, 2020). Decomposing $\Delta F_{\mathrm{aci}}^{\mathrm{eff}}$ further, Grosvenor and Carslaw (2020) found that $\Delta F_{\mathrm{aci\ N_d}}^{\mathrm{eff}}$ and $\Delta F_{\mathrm{aci\ L}}^{\mathrm{eff}}$ dominate the $\Delta F_{\mathrm{aci}}^{\mathrm{eff}}$ forcing in the northern regions of the NA (north of around 40º N), whereas $\Delta F_{\mathrm{aci\ f_c}}^{\mathrm{eff}}$ dominates further south.

   Models show that aerosol forcing has influenced the climate variability of the NA. Booth et al. (2012) showed that surface aerosol radiative forcing was the dominant driver of decadal changes in sea surface temperatures (SSTs) for the atmosphere-

ocean coupled global circulation model (the UK Met Office HadGEM2-ES model) that was used in the Fifth Coupled Model Intercomparison Project (CMIP5). Menary et al. (2020) showed that for the CMIP6 models aerosols acted to speed up the AMOC during the historical period, whereas greenhouse gases slowed it down. Climate models also predict that during the 21st Century a region in the northern NA will experience less warming under the influence of greenhouse gases than the rest of the globe (termed the NA "warming hole") related to the slowing down of the AMOC (Manabe and Stouffer, 1993; Robson

et al., 2016; Chemke et al., 2020). Over the historical period aerosols have likely delayed the formation of this warming hole by speeding up the AMOC (Dagan et al., 2020).

Despite its importance, aerosol forcing remains the most uncertain of the forcings. It would be desirable to be able to use long-term trends in observable quantities like $F_{SW\uparrow}$ to determine aerosol forcing from the observations in order to constrain models. Long-term records of $F_{SW\uparrow}$ (e.g., the Deep-C dataset for 1985-2019; Allan et al., 2014a; Liu et al., 2015, 2017) have

the potential to evaluate some aspects of model performance in terms of aerosol forcing. However, in order to understand what model evaluation using such a dataset is telling us about model performance it is necessary to understand what has been driving long-term changes in $F_{SW\uparrow}$.

There has been some previous work towards using long time records to estimate aerosol forcing and evaluate models although the feasibility of this approach remains in question. Cherian et al. (2014) used observations of surface SW flux from the GEBA

(Global Energy Balance Archive) network over Europe for the period 1990-2005 to attempt a constraint on the global aerosol forcings predicted by the CMIP5 climate models. At the locations of the GEBA stations the effective global aerosol forcings across the different models correlated with the change in surface SW model flux. The observations of the latter were then used to infer the most realistic range of effective global aerosol forcing. A potential issue with this approach is that it relies on the accuracy of the relationship between the two variables across the different models. For example, the relationship is

likely affected by the balance of forcings and feedbacks within the different models which are highly uncertain and may vary depending on the time period chosen. Kramer et al. (2021) used satellite observations to infer the total instantaneous global radiative forcing of the climate for the 2003-2018 period. This included the effect of greenhouses and a variety of other forcings, but for aerosol forcing only the ARI component was included and not ACI. Using MODIS timeseries from 2003 to 2017 for oceanic regions of the NA (off the east coast of the US and the west coast of Portugal) and off the east coast of China, Bai et al.

(2020) found no relationship between long-term changes in aerosol and changes in LWP which may indicate a forcing from aerosols via cloud adjustments that is too small to be identified over the relatively short time period given the large inter-annual variability in LWP.

One of the main complications with using long-term records to estimate aerosol forcing is that there are several other drivers of changes in clouds over long timescales that we attempt to characterize in this study. One such driver is climate change, i.e.,

changes in temperature and sea-ice cover, which causes cloud-climate feedbacks. For example, over recent decades, warming due to greenhouse gas emissions has increased rapidly, but aerosol emission rates have also varied over the historical record which will affect temperatures too. The resulting changes in clouds from cloud-climate feedbacks must be taken into consideration when attempting to estimate aerosol forcing using long-term records.

Cloud-climate feedbacks are very important in the NA region. Norris et al. (2016) showed using satellite observations that

cloud fraction has changed substantially between 1983 and 2009 and that these changes are well-predicted by models. The cloud feedback operating in this region is thought to be the mid-latitude cloud feedback whereby warming can cause an expansion of the Hadley cell and a poleward shift of the storm tracks (Held and Hou, 1980; Lu et al., 2007; Seidel et al., 2008) that can reduce mid-latitude cloudiness (Norris et al., 2016) leading to an increase in shortwave radiation reaching the surface. This amplifies the temperature change and hence is a positive feedback. Satellite observations have been used to evaluate global

model cloud feedbacks, but this approach may lead to an estimate of cloud feedback that is too negative (Armour et al., 2013; Zhou et al., 2016; Andrews et al., 2018) due to the specific pattern of SSTs that occurred over this period; namely, a cooling

over subtropical stratocumulus regions despite the overall global warming. This caused a local increase in cloud coverage over subtropical stratocumulus regions that acted to increase $F_{\text{SW}\uparrow}$ thus making the cloud feedback more negative. Care is therefore needed when using observations to infer cloud feedbacks.

In this study we use the UK Met Office climate models to better understand the underlying processes and what the observed trends in $F_{\text{SW}\uparrow}$ are telling us about model performance and the causes of climate change over the NA. There has been some work with related aims before. For example, Wang et al. (2021) showed that, across the CMIP6 models, mean cloud feedback strength and an estimate of mean aerosol forcing were negatively correlated over the 1950-2000 period such that models with a stronger negative aerosol forcing tended to have a more positive cloud feedback. This was particularly true for models that

were more consistent with the observed historical temperature change. For those models the equilibrium climate sensitivity was also negatively correlated with the aerosol forcing. These results suggest some degree of model tuning between aerosol forcing (causing a cooling) and cloud feedback (causing a warming) to allow the recreation of observed temperatures. Changes in radiative flux relative to pre-industrial times for the 1950-2000 period in the models with the strongest cloud feedback parameters were caused almost entirely by aerosol forcing rather than temperature induced feedbacks; the models with small

cloud feedback parameters showed very little change in radiative flux for this period.

We go further than the above work since we focus on simulations from one modelling centre and break down the underlying causes of the long-term shortwave radiative changes in that model in terms of clear-sky effects, cloud variables and emission types. We separate the aerosol forcing and cloud-climate feedback effects on shortwave radiative changes using different techniques to those used previously in order to more precisely estimate the aerosol forcing. Finally we also use the results

to draw conclusions about the feasibility of using long-term observations to quantify aerosol forcing and to evaluate model performance, and compare our model results to long-term observations.

## 2    Methods

### 2.1    The UKESM1 and HadGEM3-GC3.1 climate models

We examine output from the atmosphere-ocean-coupled UKESM1 (UK Earth System Model; Sellar et al., 2019) and the

HadGEM3-GC3.1 (Hadley Centre Global Environment Model 3 Global Coupled configuration version 3.1; here shortened to HadGEM; Williams et al., 2017; Kuhlbrodt et al., 2018) models, which were submitted as part of the 6[th] Coupled Model Intercomparison Project (CMIP6; Eyring et al., 2016). UKESM1 is based on the atmosphere-ocean-coupled HadGEM physical climate model, but in addition couples several Earth system processes. These additional components include the MEDUSA ocean biogeochemistry model (Yool et al., 2013), the TRIFFID dynamic vegetation model (Cox, 2001) and the stratospheric-

tropospheric version of the United Kingdom Chemistry and Aerosol (UKCA) model of atmospheric composition (Archibald et al., 2019). This version of UKCA allows a more complete description of atmospheric chemistry compared to HadGEM. For example, the latter uses an offline climatology for oxidants, whereas in UKESM1 oxidants are simulated. An N96 resolution horizontal grid is used in both models, which is $1.875 \times 1.25$ º (208 ×139 km) at the equator. 85 vertical levels are used

between the surface and 85 km altitude with a stretched grid such that the vertical resolution is 13 m near the surface and
around 150–200 m at the top of the boundary layer.

## 2.2 Model data

All CMIP6 model data originates from the Earth System Grid Federation (ESGF) archive (https://esgf-node.llnl.gov/search/cmip6/). Monthly averaged model data is used since higher time resolution data is not available for most variables. We average the monthly data to annual averages for timeseries, but use the monthly data when calculating SW fluxes (Section 3.2).

### 2.2.1 The CMIP6 UKESM1 and HadGEM coupled atmosphere-ocean ensembles

We use output from the 16-member UKESM1 and the 4-member HadGEM coupled atmosphere-ocean historical ensemble runs that were performed for CMIP6 (Sellar et al., 2019; Williams et al., 2017; Kuhlbrodt et al., 2018). These ran from 1850 to 2014 using greenhouse gas (GHG), aerosol, natural (e.g., volcanic) and other emissions that were designed to represent the real emissions over this period. We note that there are likely to be uncertainties in these emissions that will lead to model errors. The ensembles were designed to capture a range of possible ocean and atmospheric states in order to sample the natural multi-decadal variability.

### 2.2.2 The AerChemMIP and DAMIP coupled atmosphere-ocean experiments

We also make use of the DAMIP (Detection and Attribution Model Intercomparison Project; Gillett et al., 2016) and the AerChemMIP (Aerosol Chemistry Model Intercomparison Project; Collins et al., 2017) coupled atmosphere-ocean experiments to estimate the effects of individual emission types. In the HadGEM-based DAMIP experiments single sets of emission types were applied to coupled simulations. We examine DAMIP data from simulations in which only the historical anthropogenic aerosol emissions were applied (DAMIP-Hist-Aer), where only greenhouse gas emissions were applied (DAMIP-Hist-GHG) and where only natural volcanic aerosol emissions and solar forcing were applied (DAMIP-Hist-Nat). In each case the experiments are based on the same four ensemble members as for the base HadGEM experiments.

The AerChemMIP experiments are based on a 3-member subset of the 16-member UKESM1 ensemble described in Section 2.2.1, which we refer to as AerChemMIP-all-emissions. The "piAer" experiments used historical emissions for all emission types except for aerosols and aerosol pre-cursors, for which pre-industrial emissions were used. We assume that these runs are equivalent to the greenhouse gas-only runs (similar to DAMIP-Hist-GHG) since the DAMIP results (see Appendix A and B) show that aerosols and greenhouse gas emissions are the main drivers of long-term trends for the North Atlantic region. For this reason we refer to the piAer experiment as AerChemMIP-GHG-only-proxy. We estimate the effects of aerosol emissions alone by subtracting the AerChemMIP-piAer timeseries from all-emissions UKESM1 runs for the 3-member subset of ensemble members used for the AerChemMIP experiments. The accuracy of this approach is tested using the DAMIP results and is shown in Appendix B. We refer to this as AerChemMIP-aerosol-only-proxy. Box 1 of the schematic in Fig. 1 depicts the above

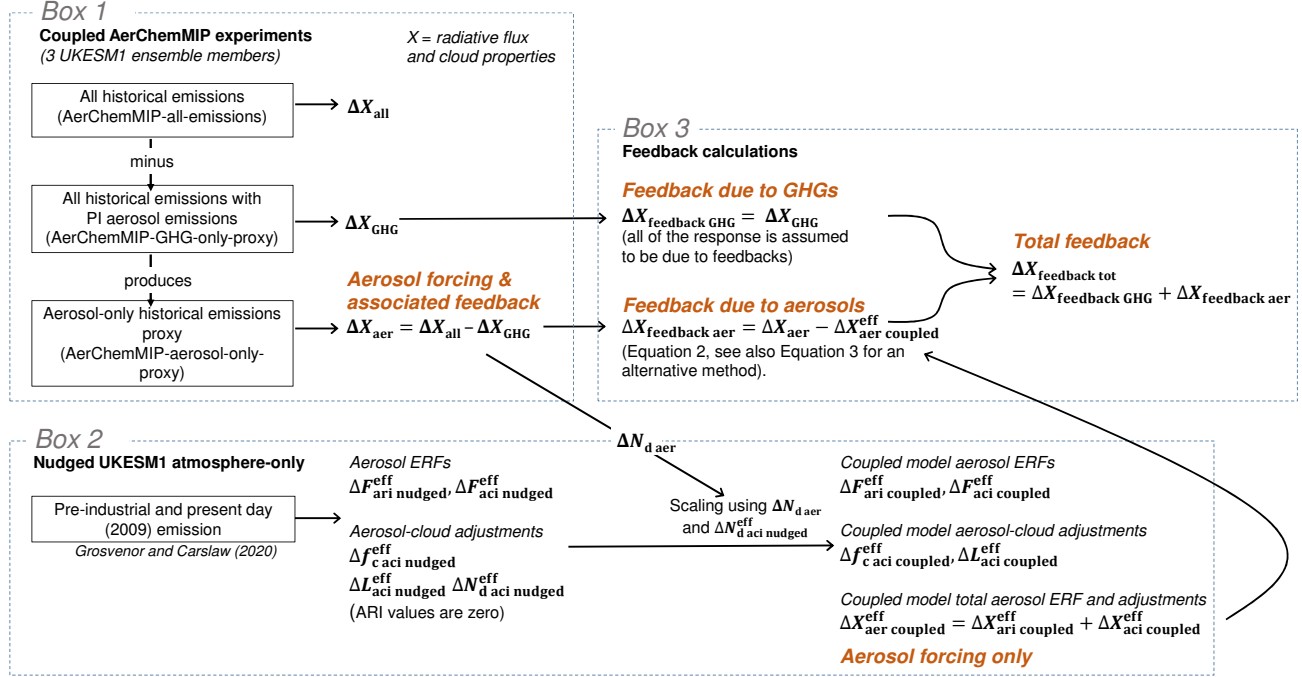

**Figure 1.** Schematic showing how various quantities are calculated. Some of the quantities are not introduced until later in the paper. The same methodology also applies to the DAMIP (HadGEM-based) results except that GHG-only and aerosol-only proxies are not required (Box 1) since there are dedicated experiments with GHG-only and aerosol-only emissions.

methodology for the AerChemMIP experiments. In the main part of the paper we focus on the UKESM1 results derived using
the AerChemMIP experiments and mostly show the DAMIP/HadGEM in the appendices.

### 2.2.3 The UKESM1 atmosphere-only run

We also examine data from the atmosphere-only UKESM1 runs performed as part of CMIP6, which have the same historical forcings as in the coupled CMIP6 runs, but with sea-surface temperatures (SSTs) and sea-ice concentrations prescribed from observations. Examination of these simulations helps to quantify how deviations of the coupled model SSTs and sea-ice from
195 the observed state affect clouds and shortwave fluxes. It also allows a model assessment of the atmospheric components against observations when given the correct ocean conditions. There is currently only one atmosphere-only run available, which prevents examination of the effects of atmospheric variability via an ensemble; for example, despite SSTs being fixed the atmosphere can exhibit different modes of variability that may not match the actual modes that occurred in reality and so some differences between the atmosphere-only run and reality might be expected.

## 2.3 Surface albedo calculation and sea-ice screening

The surface albedo ($A_{\mathrm{surf}}$) is required for the offline radiative calculations described in Section 3.2 and for the screening of high sea-ice regions. It is calculated using the monthly mean upwelling and downwelling clear-sky SW surface fluxes ($F_{\mathrm{SW\uparrow surf}}^{\mathrm{clear-sky}}$ and $F_{\mathrm{SW\downarrow surf}}^{\mathrm{clear-sky}}$, respectively) :-

$$A_{\mathrm{surf}} = \frac{F_{\mathrm{SW\uparrow surf}}^{\mathrm{clear-sky}}}{F_{\mathrm{SW\downarrow surf}}^{\mathrm{clear-sky}}} \tag{1}$$

Grid-boxes within the NA region were excluded where substantial sea-ice was formed in any of the simulations such that the same grid-boxes were excluded for all runs; the criteria for exclusion was the annual-mean surface albedo exceeding 20% at some point during the historical timeseries.

## 2.4 Uncertainties in trends

Temporal trends are calculated using a linear least squares method and the errors in the trends are calculated following Santer et al. (2000) where an effective sample size is used that takes into account temporal autocorrelation using the lag-1 autocorrelation coefficient.

## 2.5 MODIS cloud droplet number concentration observations

We use cloud droplet concentration ($N_{\mathrm{d}}$) as an indicator of aerosol-driven changes in clouds because it more directly represents the first step in the chain of processes by which aerosols affect clouds. $N_{\mathrm{d}}$ gives some indication of the number of cloud condensation nuclei (CCN; a subset of the whole aerosol population) that were available to produce clouds, but is also affected by other factors such as updraft speed, droplet collision coalesence, droplet scavenging by rain, cloud evaporation, etc.

We evaluate model $N_{\mathrm{d}}$ and its trends against MODIS $N_{\mathrm{d}}$ observations. We use a $1 \times 1^{\mathrm{o}}$ resolution data set calculated from 1 km MODIS retrievals of cloud optical depth ($\tau_{\mathrm{c}}$) and effective radius ($r_{\mathrm{e}}$). Two-dimensional fields of $N_{\mathrm{d}}$ are derived by the retrieval since it is assumed that $N_{\mathrm{d}}$ is constant throughout the depth of the cloud, which has been shown to be a good approximation by aircraft observations of stratocumulus (Painemal and Zuidema, 2011). Details of the retrieval and dataset are given in Grosvenor and Carslaw (2020). For the model, two-dimensional $N_{\mathrm{d}}$ fields were obtained from the monthly 3D $N_{\mathrm{d}}$ fields by calculating a weighted vertical mean $N_{\mathrm{d}}$, with the liquid water mixing ratio ($q_{\mathrm{L}}$) on each level used for the weights. This ensures that the levels with the most $q_{\mathrm{L}}$ contribute most to the average $N_{\mathrm{d}}$, which is similar to what is assumed in the MODIS retrieval since most of the $r_{\mathrm{e}}$ signal comes from near cloud top where $q_{\mathrm{L}}$ is usually the largest and the $N_{\mathrm{d}}$ calculation is a strong function of $r_{\mathrm{e}}$. It also reduces the weight of contributions from very thin clouds that would not be detected by MODIS. Only datapoints for which the mean cloud top height is below 3.2 km were included for the satellite $N_{\mathrm{d}}$ calculation in order to help exclude satellite retrieval errors for deeper clouds (see Grosvenor et al., 2018).

## 2.6 Variables considered and assumptions for changes in shortwave flux

We attribute trends in $F_{\text{SW}\uparrow}$ to changes in liquid clouds, clear-sky $F_{\text{SW}\uparrow}$ and surface albedo. Changes in clear-sky $F_{\text{SW}\uparrow}$
($F_{\text{SW}\uparrow}^{\text{clear}-\text{sky}}$) will include the effects of changes in aerosol in cloud-free air, changes in the surface albedo ($A_{\text{surf}}$) and changes in trace gas concentrations. However, we do not attempt to separate these effects here. For changes in the all-sky (i.e., combined cloudy and clear regions) albedo, we consider the effect of changes in the three main variables that affect it, namely cloud fraction ($f_{\text{c}}$), cloud droplet number concentration ($N_{\text{d}}$) and in-cloud liquid water path ($L$), along with $F_{\text{SW}\uparrow}$ from the clear-sky regions above clouds and also $A_{\text{surf}}$ in cloudy-sky conditions. However, changes in the latter were found to have negligible impact for the region considered. $L$ is the LWP from the cloudy regions only. For $f_{\text{c}}$ we use the total cloud fraction since liquid only cloud fractions aggregated over all heights (i.e., accounting for overlap assumptions) were not available. Occasionally, the all-sky liquid water path ($L_{\text{all}-\text{sky}}$) is also considered (i.e., including both the cloudy and clear-sky portions of model gridboxes or observed regions). To calculate $L$ from the $L_{\text{all}-\text{sky}}$ values provided by the models we assume that $L = L_{\text{all}-\text{sky}}/f_{\text{c}}$ (e.g., as also in Seethala and Horváth, 2010); we use monthly values for this calculation.

## 2.7 Calculation of aerosol radiative forcing

The effective radiative forcings (ERFs) due to aerosol-cloud interactions ($\Delta F_{\text{aci}}^{\text{eff}}$) and aerosol-radiation interactions ($\Delta F_{\text{ari}}^{\text{eff}}$) are considered. The total aerosol ERF ($\Delta F_{\text{aer}}^{\text{eff}}$) is the sum of the two. In the coupled climate runs SSTs vary over time and so ERFs cannot be directly calculated from the change in $F_{\text{SW}\uparrow}$ in aerosol-only emissions runs. Instead, the ERFs for the coupled climate runs (see Section 3.5) were estimated by scaling the $\Delta F_{\text{ari}}^{\text{eff}}$ and $\Delta F_{\text{aci}}^{\text{eff}}$ from nudged simulations based on the ratio of the change in $N_{\text{d}}$ over time in the coupled models to the change in $N_{\text{d}}$ in the nudged runs (see Box 2 of Fig 1; Appendix C gives more details of the calculations). The nudged model runs consist of a pair of atmosphere-only, nudged UKESM1 simulations with prescribed time-varying SSTs, as presented in Grosvenor and Carslaw (2020); one simulation used pre-industrial (PI) aerosol emissions and the other present day (PD) emissions from 2009. The nudging (using 2009 reanalysis) was applied only to the winds above the boundary layer and kept the large scale meteorology approximately the same in both simulations whilst allowing local boundary layer and associated clouds to respond to the different aerosol loadings.

## 3 Results

### 3.1 North Atlantic timeseries for UKESM1

Figure 2 shows UKESM1 and HadGEM timeseries averaged over a region of the North Atlantic (defined by the black box in Fig. 3 for ocean grid-boxes with no substantial sea-ice, see Section 2.3). The ensemble mean $F_{\text{SW}\uparrow}$ shows two long-term trends. The first is a positive trend between 1850 and approximately 1970; we denote this time period as pre-1970. The second is a negative trend from 1971 to the end of the simulation in 2014; denoted post-1970. $F_{\text{SW}\uparrow}$ in 2014 and 1850 are similar despite the atmosphere not being free from anthropogenic influence at this time. The reasons for the similarity are discussed later.

For each variable, trends have been fitted to the ensemble mean timeseries for the two periods and then multiplied by the
duration of the time periods to give the total change in quantity x (denoted $\Delta$x) .These values and the associated uncertainties
in the fitted trends are given in Table 1. For the pre-1970 period for UKESM1 $\Delta F_{\text{SW}\uparrow}$ was 4.7$\pm$0.98 W m$^{-2}$ and over the
post-1970 period it was -6.0$\pm$1.4 W m$^{-2}$, hence the magnitude of the change in $F_{\text{SW}\uparrow}$ is slightly larger for the second period.
Short periods of enhanced $F_{\text{SW}\uparrow}$ are evident that reach close to, or extend beyond, the 2-sigma variation of the ensemble. These
are due to volcanic eruptions. One example occurs in 1991 and is due to the Mount Pinatubo eruption.

The maps of $\Delta F_{\text{SW}\uparrow}$ in Fig. 3 show that the NA is one of the main oceanic regions that show large changes in $F_{\text{SW}\uparrow}$ over
the chosen time periods, which justifies the choice of this region as the focus of this paper. The other oceanic regions that
show large changes are the Barents Sea (north of Scandinavia), the Southern Ocean and the northern Pacific. The Barents Sea
and Southern Ocean changes in $F_{\text{SW}\uparrow}$ are likely to be related to sea-ice changes. The North American and Western European
continental regions also show large changes that are often larger than those over the ocean regions. For the pre-1970 period the
UKESM and HadGEM models are consistent in that larger changes in $F_{\text{SW}\uparrow}$ occur in the western North Atlantic region than
in the east suggesting a connection with pollution outflow from North America. This is also true for the post-1970 period for
the HadGEM model, but for the UKESM model there is a stronger change in the eastern part of the North Atlantic suggesting
that different processes may be occurring compared to pre-1970 or potentially more natural variability in the spatial patterns.

We now discuss the potential drivers of changes in $F_{\text{SW}\uparrow}$. Cloud fraction shows a small increase over the pre-1970 period
($\Delta f_{\text{c}} = (8.4\pm2.9)\times10^{-3}$) whereas over the post-1970 period there is a distinct decrease ($\Delta f_{\text{c}} = (-33.9\pm3.7)\times10^{-3}$). The start
of the negative trend in cloud fraction occurs at around the same time as the start of the negative trend in $F_{\text{SW}\uparrow}$ (1971). $N_{\text{d}}$
shows a large increase over the pre-1970 period ($\Delta N_{\text{d}} = 46.3\pm8.7$ cm$^{-3}$) and, similarly to $F_{\text{SW}\uparrow}$ and $f_{\text{c}}$, decreases ($\Delta N_{\text{d}}$
$= -20.2\pm6.4$ cm$^{-3}$) after around 1971. Aerosol optical depth at 550nm ($\tau_{\text{a}}$, including dust) shows very similar trends to $N_{\text{d}}$
although it is more variable. $L$ increases over the pre-1970 period ($\Delta L = 3.8\pm0.40$ g m$^{-2}$) indicating cloud thickening but
shows very little change over the post-1970 period ($\Delta L = -0.81\pm0.60$ g m$^{-2}$). The reasons for the changes in the different
cloud variables are discussed in Section 3.3.2.

The clear-sky TOA radiative flux ($F_{\text{SW}\uparrow}^{\text{clear}-\text{sky}}$) also increases over the pre-1970 period ($\Delta F_{\text{SW}\uparrow}^{\text{clear}-\text{sky}}=1.6\pm0.85$ W m$^{-2}$) and
decreases thereafter. $\Delta F_{\text{SW}\uparrow}^{\text{clear}-\text{sky}}$ over the post-1970 period (-1.6$\pm$1.8 W m$^{-2}$) is the same magnitude, but of opposite sign,
to that over the pre-1970 period. There are also large spikes in $F_{\text{SW}\uparrow}^{\text{clear}-\text{sky}}$ due to volcanic eruptions which are not present in
the cloud variables suggesting that the effect of volcanoes on $F_{\text{SW}\uparrow}$ occurs mainly via clear-sky effects. Note that the clear-
sky effects are likely to be negligible in the cloudy parts of the grid boxes; hence the $\Delta F_{\text{SW}\uparrow}^{\text{clear}-\text{sky}}$ values would need to be
multiplied by the clear-sky fraction (=1-$f_{\text{c}}$) to give the clear-sky contribution to the overall $\Delta F_{\text{SW}\uparrow}$.

On the whole the changes in variables and trends in HadGEM model are very similar to those for UKESM1 although with
slightly smaller magnitude changes in $F_{\text{SW}\uparrow}$, $N_{\text{d}}$ and $L$ and larger magnitude changes in $\tau_{\text{a}}$ (see Table A1 and Appendix A for
details on the HadGEM results). In addition there is a notable difference in the magnitude of $F_{\text{SW}\uparrow}^{\text{clear}-\text{sky}}$ with HadGEM being
around 1 W m$^{-2}$ higher than UKESM1, which is consistent with the higher $\tau_{\text{a}}$ values. The reasons for this are left to other
work to explore. Due to similarity of the two models we mostly focus on the UKESM1 model for this paper and show results
from HadGEM in Appendix A.

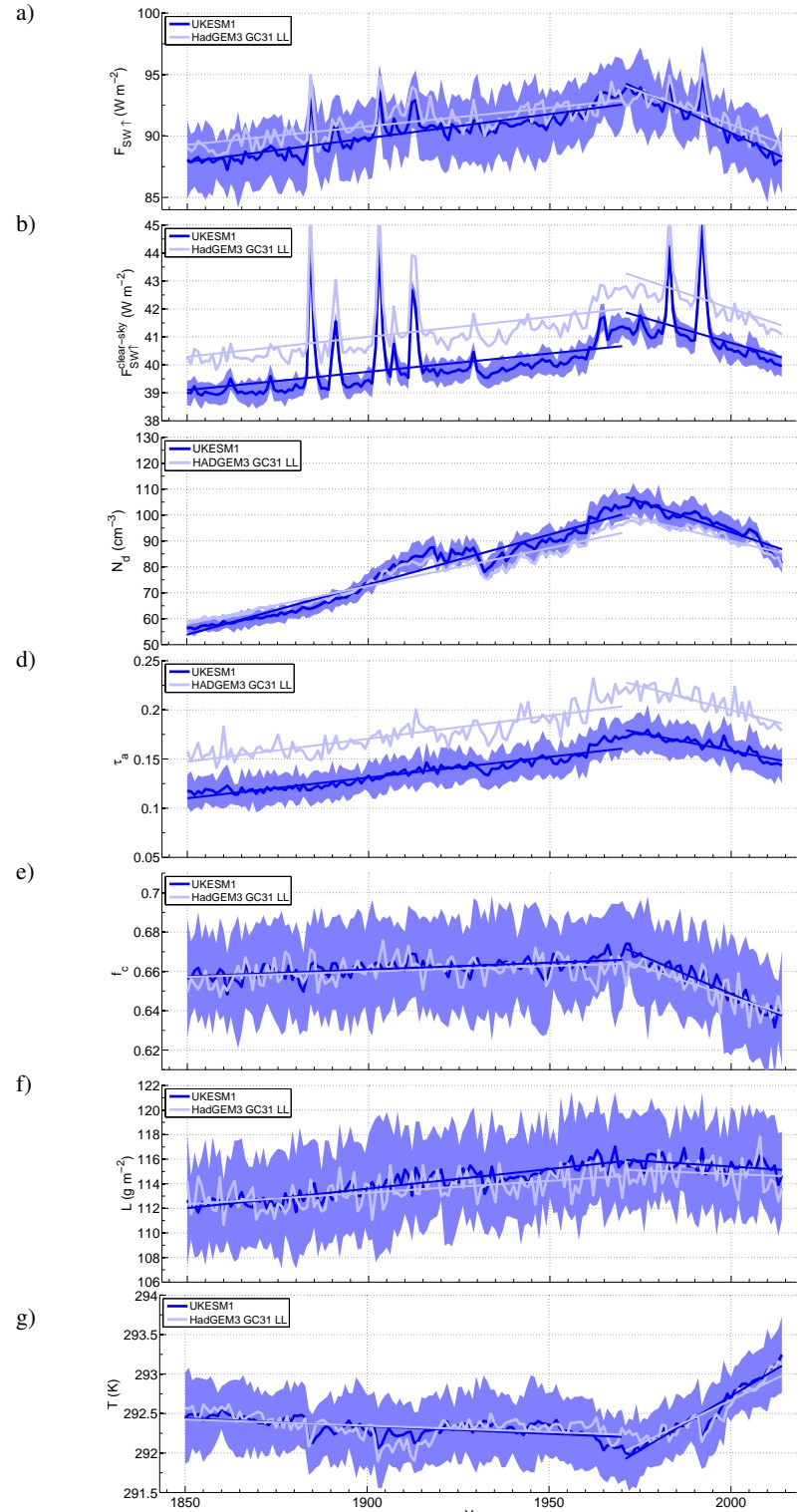

**Figure 2.** Timeseries of annual mean values of various quantities from the UKESM1 model spatially averaged over the North Atlantic region (18-60° N, 0-75° W; ocean-only gridpoints with sea-ice regions excluded, see text for details). The blue shading denotes ±2 times the standard deviation across the ensemble for UKESM1 only. a) the all-sky upwelling SW flux at TOA ($F_{\mathrm{SW\uparrow}}$); b) the all-sky upwelling SW flux at TOA ($F_{\mathrm{SW\uparrow}}^{\mathrm{clear-sky}}$); c) The vertically averaged (weighted by liquid water content) cloud droplet concentration ($N_{\mathrm{d}}$); d) the aerosol+dust

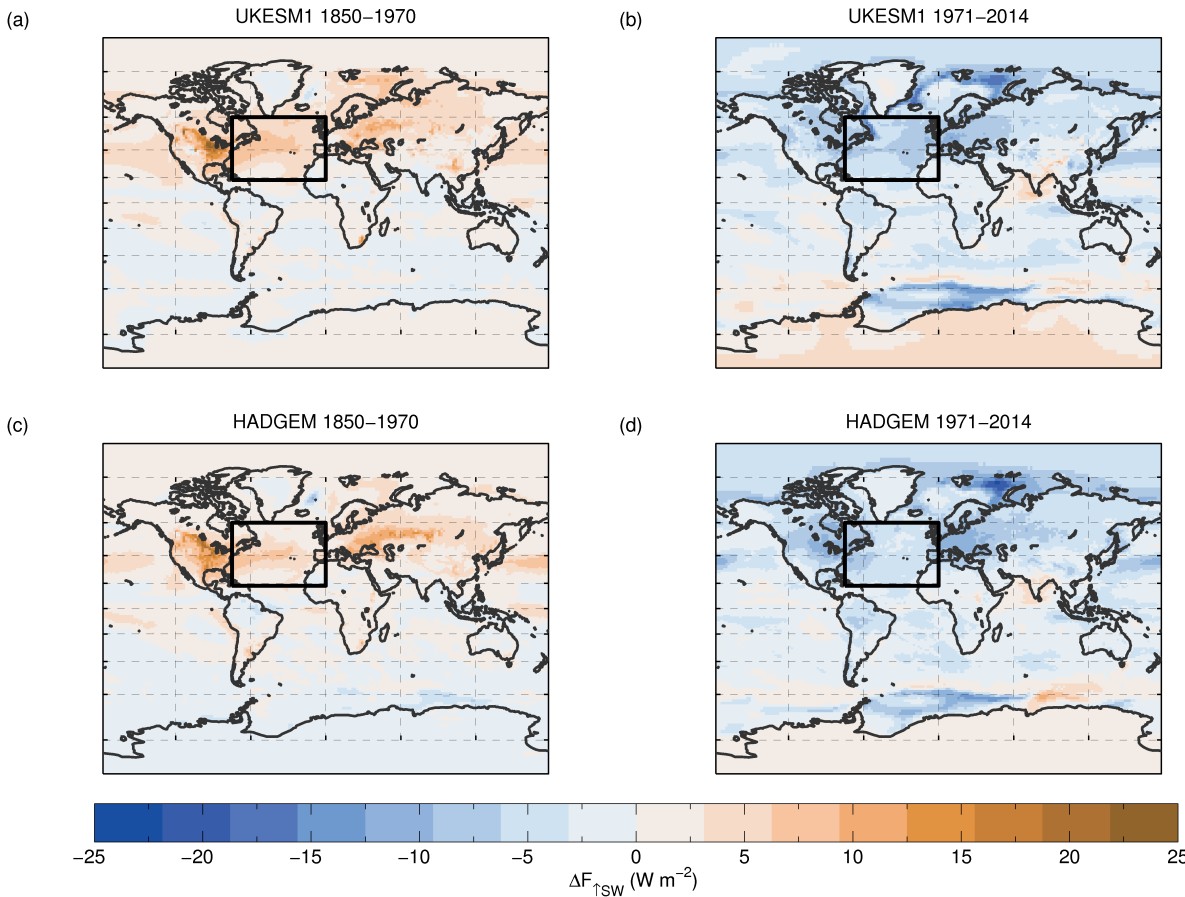

**Figure 3.** Maps of the change in $F_{SW\uparrow}$ over the 1850-1970 (left column) and 1971-2014 (right column) periods for the ensemble means of the UKESM1 (top row) and HadGEM (bottom row) models. The region used for the timeseries analysis is shown by a black box; note that land regions and regions with substantial sea-ice within the box were excluded (see text for details).

**Table 1.** Changes in radiative fluxes and cloud properties ($\Delta_x$ values) over the 1850-1970 and 1971-2014 periods for the ensemble mean timeseries of the UKESM1 and AerChemMIP simulations. Also shown are the minimum and maximum $\Delta_x$ values across the UKESM1 16-member ensemble. The uncertainties are from the uncertainty in the fit lines used to calculate $\Delta_x$.

| Period | | $\Delta F_{\text{SW}\uparrow}$ $(Wm^{-2})$ | $\Delta F_{\text{SW}\uparrow}^{\text{clear}-\text{sky}}$ $(Wm^{-2})$ | $\Delta f_c$ (x $10^{-3}$) | $\Delta N_d$ $(cm^{-3})$ | $\Delta \tau_a$ (x $10^{-2}$) | $\Delta L$ $(gm^{-2})$ | $\Delta T$ (K) |
|---|---|---|---|---|---|---|---|---|
| 1850-1970 | UKESM1 ensemble mean | 4.7±0.98 | 1.6±0.85 | 8.4±2.9 | 46.3±8.7 | 4.9 ± 0.9 | 3.8±0.40 | -0.23± 0.16 |
| | UKESM1 min ensemble trend | 2.7± 1.3 | 1.5± 0.81 | -8.5± 9.3 | 44.8± 5.8 | 4.7 ± 0.5 | 2.8± 1.4 | -0.60± 0.34 |
| | UKESM1 max ensemble trend | 5.9±1.0 | 1.7±0.85 | 17.1±8.9 | 47.4±6.0 | 5.2 ± 0.8 | 5.5±1.3 | 0.24±0.24 |
| | AerChemMIP-aerosol-only-proxy | 6.6±0.81 | 1.8±0.81 | 22.2±7.5 | 47.7±4.7 | 5.1 ± 0.7 | 5.3±1.0 | -0.82±0.17 |
| | AerChemMIP-GHG-only-proxy | -2.3±0.73 | -0.25±0.73 | -17.3±5.9 | -1.0±0.49 | -0.02 ± 0.35 | -1.00±0.77 | 0.68±0.14 |
| 1971-2014 | UKESM1 ensemble mean | -6.0±1.4 | -1.6±1.8 | -33.9±3.7 | -20.2±6.4 | -3.0 ± 0.6 | -0.81± 0.60 | 1.2± 0.23 |
| | UKESM1 min ensemble trend | -7.6± 1.4 | -1.8± 1.9 | -46.4± 14.9 | -21.8± 5.0 | -3.6 ± 0.7 | -2.5± 2.2 | 0.88± 0.26 |
| | UKESM1 max ensemble trend | -4.9±1.8 | -1.5±1.8 | -25.6±12.0 | -16.0±6.1 | -2.7 ± 0.5 | 0.62±2.2 | 1.6±0.19 |
| | AerChemMIP-aerosol-only-proxy | -3.4±1.1 | -1.2±1.1 | -9.8±10.4 | -20.9±6.7 | -3.5 ± 1.3 | -1.3±2.1 | 0.43±0.12 |
| | AerChemMIP-GHG-only-proxy | -2.8±1.8 | -0.53±1.8 | -23.4±9.8 | 0.54±0.91 | 0.1 ± 0.5 | -0.07±1.2 | 0.80±0.25 |

## 3.2 Decomposing the $F_{\text{SW}\uparrow}$ trends in UKESM1 into contributions from individual variables

The above results show that the increase in $F_{\text{SW}\uparrow}$ over the pre-1970 period is likely to be caused by a combination of increases in $N_d$, $L$ and $F_{\text{SW}\uparrow}^{\text{clear}-\text{sky}}$ since there is little change in $f_c$. In contrast, for the post-1970 period the $F_{\text{SW}\uparrow}$ decrease is likely to be caused by decreases in $f_c$, $N_d$ and $F_{\text{SW}\uparrow}^{\text{clear}-\text{sky}}$ since $L$ is fairly constant. To quantify the relative contributions of the changes in cloud properties to the changes in $F_{\text{SW}\uparrow}$ we first recreate the $F_{\text{SW}\uparrow}$ flux timeseries using offline radiative flux calculations with monthly ensemble mean $f_c$, $N_d$, $L$, $F_{\text{SW}\uparrow}^{\text{clear}-\text{sky}}$, downwelling SW at TOA and $A_{\text{surf}}$ as inputs following the technique described in Grosvenor et al. (2017) and Grosvenor and Carslaw (2020) for TOA fluxes. The approach used here differs slightly to those studies due to the inclusion here of $F_{\text{SW}\uparrow}^{\text{clear}-\text{sky}}$ from the model for the clear-sky regions rather than assuming a constant transmissivity. Multiple scattering between the surface and cloud is also included here following Seinfeld and Pandis (2006). The offline radiative flux calculations can then be used to quantify the individual contributions from the changes in the different cloud properties.

Figure 4a compares the reconstructed $F_{\text{SW}\uparrow}$ flux timeseries with the timeseries from the model (i.e., that calculated online by the UKESM1 at each radiation timestep of the model, as previously shown in Fig 2). The inter-annual variability of the calculated fluxes match those from the model output very well. The $\Delta F_{\text{SW}\uparrow}$ values from the reconstructed timeseries are similar to the actual model values during the pre-1970 period and the post-1970 period (see Table 2), although with a 6% overestimate for the pre-1970 period ($\Delta F_{\text{SW}\uparrow}$ = 5.0 vs 4.7 W m$^{-2}$ for the estimated vs actual values, respectively), and a 20% underestimate in the absolute magnitude of $\Delta F_{\text{SW}\uparrow}$ for the post-1970 period ($\Delta F_{\text{SW}\uparrow}$ = -4.8 vs -6.0 W m$^{-2}$). Despite these discrepancies, the appearance of a positive trend in the pre-1970 period and a negative trend in the post-1970 period, along with trends that are close to those from the full model gives confidence that the reconstructed radiative fluxes are sufficient for estimating the contributions from the individual cloud properties to $\Delta F_{\text{SW}\uparrow}$.

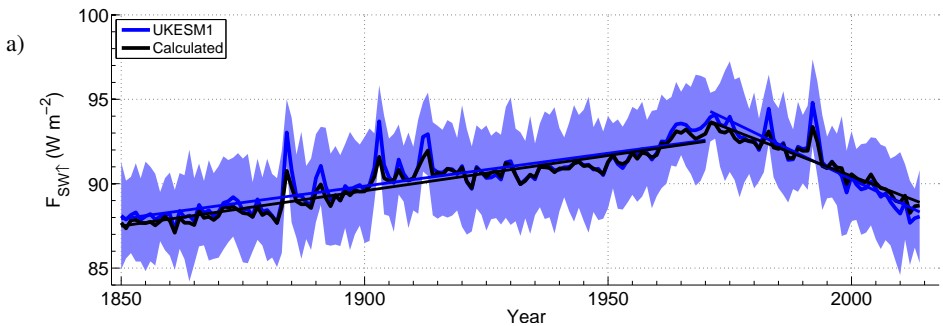

**Figure 4.** Timeseries of annual mean $F_{SW\uparrow}$ as calculated using the offline radiative transfer model (labelled "Calculated") and that directly from the model output for the UKESM1. The blue shading denotes $\pm 2$ times the standard deviation across the ensemble for the model output data. The region for area averaging is the same as for Fig. 2.

The individual contributions to $\Delta F_{SW\uparrow}$ were estimated by re-calculating the $F_{SW\uparrow}$ fluxes, and the linear trends in $\Delta F_{SW\uparrow}$, for the two periods while holding the other cloud properties fixed at the time-mean value for each time period.

For the pre-1970 period, all variables cause an increase in $F_{SW\uparrow}$ trend. The trend in $N_d$ contributes most to $\Delta F_{SW\uparrow}$ (Figure 5 and Table 2) with 58.6% of the total, followed by $L$ (20.0%), then $f_c$ (13.8%) and $F_{SW\uparrow}^{clear-sky}$ (9.5%), with a -2.6% residual. For the post-1970 period the largest influence is the reduction in $f_c$ which explains 64.8% of $\Delta F_{SW\uparrow}$. However, the decrease in $N_d$ also has some influence (20.4%). $F_{SW\uparrow}^{clear-sky}$ and $L$ decrease slightly over this period, but have minimal influence on the $F_{SW\uparrow}$ change (10.1% and 7.5%), respectively, and with large uncertainties. There is a small residual of -0.57%.

These results show that the long-term changes in $F_{SW\uparrow}$ over the pre-1970 period are dominated by cloud brightening via the Twomey effect (i.e., an increase in $N_d$ with other cloud properties held constant). The increase in the macrophysical cloud properties, $L$ and $f_c$, which account for a combined 33.8% of $\Delta F_{SW\uparrow}$, could indicate some cloud adjustments in response to changes in $N_d$, but could also be influenced by non-aerosol factors such as changes in SST, air temperature, or atmospheric and oceanic circulation. These effects will be discussed in the next section. For the post-1970 period the Twomey effect (-0.98 W m$^{-2}$) is considerably smaller in magnitude than for the pre-1970 period (2.9 W m$^{-2}$) because $\Delta N_d$ is only -20.2$\pm$6.4 cm$^{-3}$ in the post-1970 period compared to 46.3$\pm$8.7 cm$^{-3}$ in the pre-1970 period. Another factor is that cloud albedo, and hence $F_{SW\uparrow}$, is more sensitive to changes in $N_d$ when $N_d$ is lower (Carslaw et al., 2013), so the reduction in $N_d$ between its peak in 1971 and 2014 will have had less effect on $F_{SW\uparrow}$ compared to the same $\Delta N_d$ in the pre-industrial-like conditions of 1850; $\Delta F_{SW\uparrow}$ for the post-1970 period is 34% of the pre-1970 value, whereas the post-1970 $\Delta N_d$ is 44% of the pre-1970 value. The much larger change in $f_c$ during the post-1970 period compared to the pre-1970 period suggests that the reduction in $f_c$ is unlikely to be dominated by cloud adjustments to aerosol given that the change in $N_d$ is much smaller over the post-1970 period than over the pre-1970 period. There are several factors that could influence the macrophysical cloud changes during the two periods and we now attempt to quantify the influence of the individual drivers.

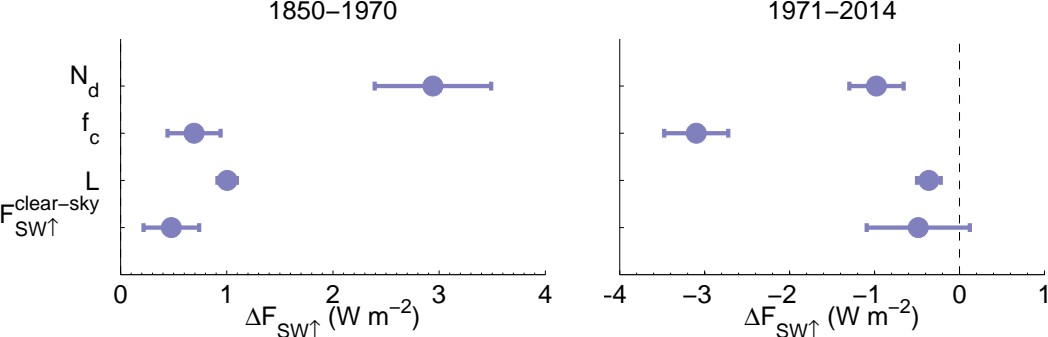

**Figure 5.** Estimated contributions to the change in $F_{SW\uparrow}$ ($\Delta F_{SW\uparrow}$) between 1870 and 1970 (the pre-1970 period) and between 1971 and 2014 (the post-1970 period) due to changes in $N_d$, $f_c$, $L$ and $F_{SW\uparrow}^{clear-sky}$ calculated using an offline radiative transfer algorithm by allowing only one variable at a time to vary. All results are for the UKESM1 model.

**Table 2.** $\Delta F_{SW\uparrow}$ values for the two periods. Shown are the actual values from the model output (online radiative calculations), the reconstructed values (using offline radiative calculations with all variables changing over time) and the estimated contributions from changes in $f_c$, $N_d$, $L$, clear-sky $F_{SW\uparrow}$ ($F_{SW\uparrow}^{clear-sky}$) and surface albedo. N.B., the surface albedo contribution here only includes that from cloudy conditions; the effect of surface albedo in clear skies is included in $F_{SW\uparrow}^{clear-sky}$. The contributions are also quoted as percentages of the full offline values (offline radiative calculations with all variables changing over time). Residuals from the the full offline values are also listed.

| | UKESM1 | | AerChemMIP-aerosol-only-proxy | | AerChemMIP-GHG-only-proxy | |
|---|---|---|---|---|---|---|
| | 1850-1970 | 1971-2014 | 1850-1970 | 1971-2014 | 1850-1970 | 1971-2014 |
| Actual $\Delta F_{SW\uparrow}$ (W m$^{-2}$) | 4.7±0.98 | -6.0±1.4 | 6.6±1.7 | -3.4±3.3 | -2.3±0.73 | -2.8±1.8 |
| Offline $\Delta F_{SW\uparrow}$ (W m$^{-2}$) | 5.0±0.70 | -4.8±0.88 | 6.7±1.2 | -2.7±1.9 | -1.9±0.49 | -2.2±1.1 |
| Contribution to $\Delta F_{SW\uparrow}$ (W m$^{-2}$) from:- | | | | | | |
| $f_c$ ($\Delta F_{aci\,f_c}^{eff}$) | 0.69±0.25 (13.8%) | -3.1±0.38 (64.8%) | 1.9±1.1 (27.8%) | -1.1±1.6 (39.6%) | -1.5±0.54 (77.2%) | -2.0±0.84 (90.5%) |
| $N_d$ ($\Delta F_{aci\,N_d}^{eff}$) | 2.9±0.55 (58.6%) | -0.98±0.32 (20.4%) | 3.0±0.36 (44.2%) | -0.95±0.38 (34.7%) | -0.07±0.04 (3.7%) | -0.01±0.07 (0.31%) |
| $L$ ($\Delta F_{aci\,L}^{eff}$) | 1.0±0.09 (20.0%) | -0.36±0.14 (7.5%) | 1.5±0.35 (21.8%) | -0.41±0.67 (15.1%) | -0.34±0.19 (17.6%) | -0.14±0.28 (6.4%) |
| Clear-sky | 0.48±0.26 (9.5%) | -0.48±0.61 (10.1%) | 0.53±0.53 (7.8%) | -0.33±1.3 (12.1%) | -0.03±0.27 (1.8%) | -0.20±0.66 (9.2%) |
| Surface albedo | 0.03±0.03 (0.66%) | -0.01±0.08 (0.28%) | -0.00±0.07 (-0.04%) | -0.00±0.16 (0.04%) | 0.04±0.03 (-1.8%) | -0.01±0.08 (0.58%) |
| Residual | -0.13 (-2.6%) | 0.15 (-0.57%) | -0.03 (1.5%) | 0.15 (-4.2%) | -0.11 (-1.6%) | 0.04 (-4.4%) |

### 3.3 Quantifying the effects of individual emission types on $F_{SW\uparrow}$ and cloud variable changes

So far we have attributed the changes in $\Delta F_{SW\uparrow}$ to changes in cloud and aerosol properties. We now attempt to attribute the changes in radiative fluxes and the associated cloud variables to changes in emissions (see Section 2.2.2) in UKESM1, based on the AerChemMIP experiments, and in HadGEM (see Appendix A), based on the DAMIP experiments. We do this for several variables ($F_{SW\uparrow}$, $F_{SW\uparrow}^{clear-sky}$, $N_d$, $\tau_a$, $f_c$, $L$ and surface temperature) by fitting trends to the AerChemMIP and DAMIP timeseries for the pre-1970 period and the post-1970 periods and calculating the change in the trend lines as a $\Delta x$ value as described in Section 3.1. The values are listed in Table 1.

#### 3.3.1 Effect of emissions on $F_{SW\uparrow}$ and $F_{SW\uparrow}^{clear-sky}$

Fig. 6 shows the timeseries of $F_{SW\uparrow}$ and the cloud variables expressed as an anomaly relative to the 1850-1859 mean for the AerChemMIP aerosol-only and greenhouse gas-only proxies (see Section 2.2.2). Anthropogenic aerosol emissions (AerChemMIP-aerosol-only-proxy) generally cause an increase in $F_{SW\uparrow}$ whereas greenhouse gas emissions (AerChemMIP-GHG-only-proxy) cause a decrease. When all emissions are applied (AerChemMIP-all-emissions) the effects of aerosols and greenhouse gases act in opposite directions resulting in a smaller magnitude change in $F_{SW\uparrow}$ than would occur with only one of the emission types. For the majority of the timeseries, changes in aerosols have the most influence, therefore there is an overall increase in $F_{SW\uparrow}$ over most of the timeseries. However, by the end of the timeseries, $F_{SW\uparrow}$ is similar to the value at the start.

Figure 7 and Table 1 summarize the contributions of each emission type to $\Delta F_{SW\uparrow}$ in UKESM1. For the pre-1970 period the $\Delta F_{SW\uparrow}$ estimated to be due to aerosol emissions is 6.6±1.7 W m$^{-2}$ (see Table 1), which is much larger in magnitude than the reduction in $F_{SW\uparrow}$ caused by greenhouse gas emissions (-2.3±0.73 W m$^{-2}$). However, the reduction due to greenhouse gases is still important and shows that in the models with all emissions applied the effect of SW aerosol forcing is offset by around 35% by opposing greenhouse gas effects. For the post-1970 period there is less contribution from aerosol emissions (-3.4±3.3 W m$^{-2}$), which is consistent with the smaller magnitude change in $N_d$ due to aerosol emission reductions (-20.9±6.7 vs 47.7±4.7 cm$^{-3}$ for the pre-1970 period). There is a similarly-sized negative contribution from greenhouse gas emissions (-2.8±1.8 W m$^{-2}$).

For $F_{SW\uparrow}^{clear-sky}$ only the aerosol emissions drive meaningful trends suggesting that greenhouse gas driven changes in clear-sky SW are negligible (e.g., those caused by changes in water vapour or gaseous absorption).

#### 3.3.2 Effect of emissions on $f_c$, $N_d$, $L$ and $\tau_a$

We next consider how the individual emission types affect the underlying cloud variables that were shown in the previous sections to drive the changes in $F_{SW\uparrow}$. Fig. 7 shows the overall changes in $f_c$, $N_d$ and $L$ for AerChemMIP-all-emissions, AerChemMIP-aerosol-only-proxy and AerChemMIP-GHG-only-proxy.

Fig. 7 shows that the magnitude of the greenhouse gas-driven decrease in $f_c$ is slightly larger in the post-1970 period than in the pre-1970 period. Aerosols cause a positive $\Delta f_c$ in the pre-1970 period and a slightly negative $\Delta f_c$ in the post-1970

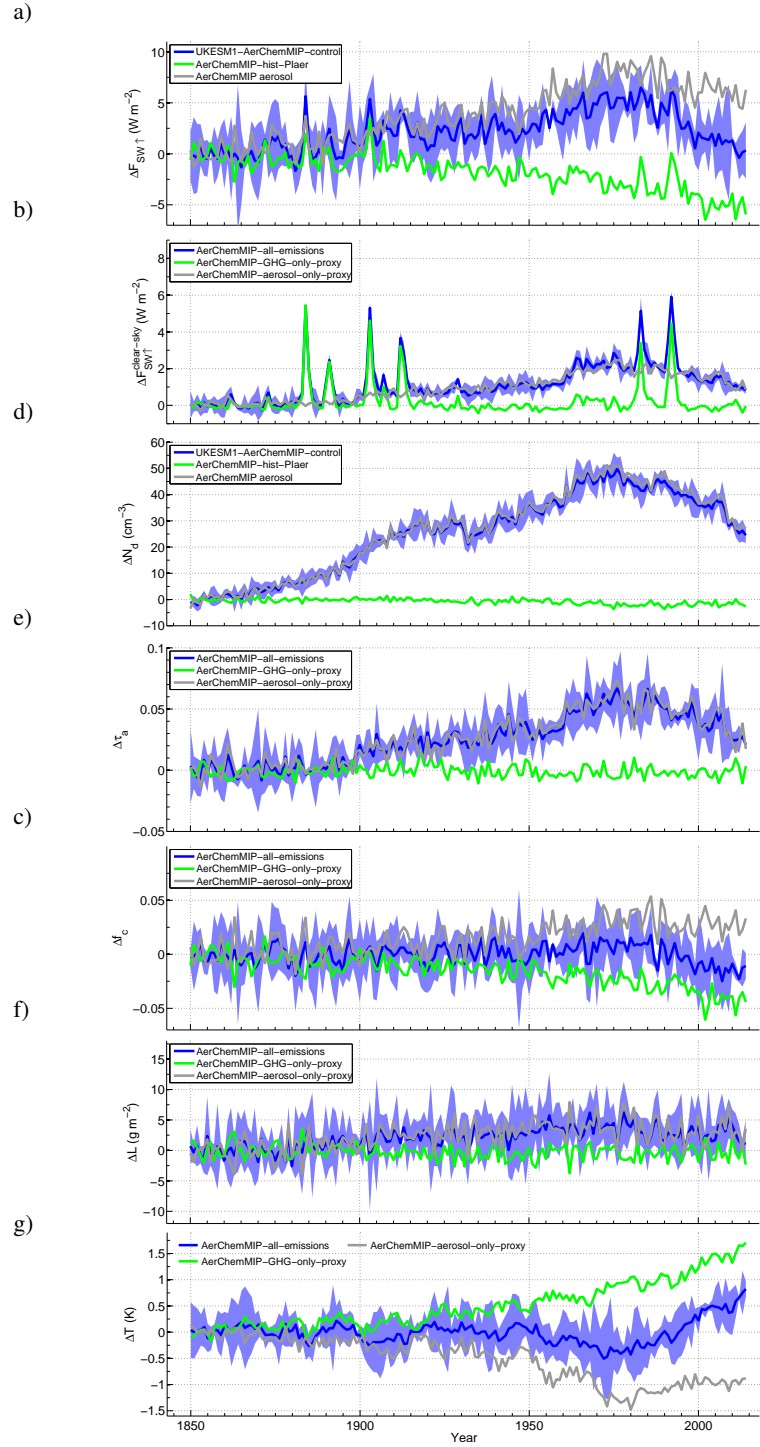

**Figure 6.** As for Fig. 2 except for the various single-emission AerChemMIP proxy simulations and that values are expressed as a perturbation from the average over the first 10 years of simulation for each line. Lines are shown for AerChemMIP-all-emissions, AerChemMIP-aerosol-only-proxy and AerChemMIP-GHG-only-proxy.

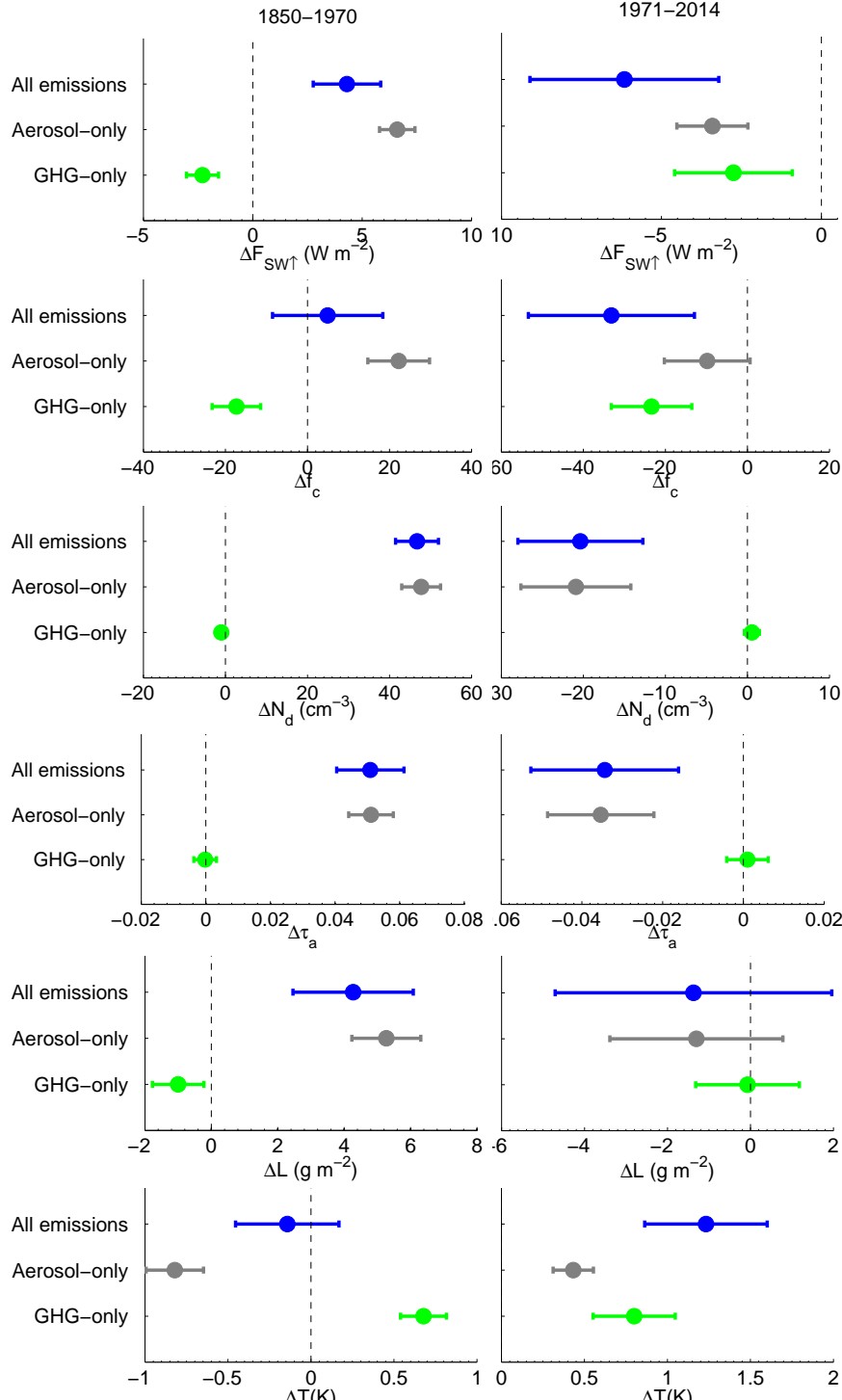

**Figure 7.** Changes in various quantities for the 1850-1970 period (left column) and the 1971-2014 period (right column) for the AerChemMIP UKESM1 experiments. Results are shown for the AerChemMIP-GHG-only-proxy, AerChemMIP-aerosol-only-proxy and AerChemMIP-all-emissions experiments.

period. Fig. 5 showed that in the pre-1970 period for the UKESM1 run there is little net contribution to $\Delta F_{SW\uparrow}$ from changes in $f_c$ with changes in $N_d$ dominating. The results for the UKESM all-emissions run in Figs. 6 and 7 show that this is due to a fairly small net change in $f_c$ during this period for UKESM1 relative to the post-1970 period. However, the AerChemMIP experiments suggest that this small change in $f_c$ is the result of opposing large changes due to the aerosol and greenhouse gas emissions.

Changes in $N_d$ (and $\tau_a$) are dominated by the aerosol emissions during both periods with virtually no contribution from greenhouse gases. This indicates that the substantial changes to climate from greenhouse gases have no effect on $N_d$ or aerosols in this model. It is conceivable that changes in cloud location, cloud coverage, updraft speeds or precipitation in response to greenhouse gases could affect $N_d$, but this appears not to be the case for this model.

The dominant driver of $\Delta L$ (Fig. 6 and 7) during the pre-1970 period is aerosol emissions (AerChemMIP-aerosol-only-proxy) and there is no significant change in $L$ due to greenhouse gas emissions. During the post-1970 period contributions to $\Delta L$ from greenhouse gases are near zero and there is a small negative aerosol contribution. However, the uncertainties for this period are larger than the values indicating that they are likely spurious.

### 3.3.3 Effect of emissions on surface temperature

During the pre-1970 period the warming from greenhouse gases (0.68±0.14 K) and the cooling from aerosols (-0.82±0.17 K) roughly cancel out to give a net change in temperature that is nearly zero (-0.14±0.19 K). During the post-1970 period greenhouse gases produce a warming of 0.80±0.25 K that is similar to that for the pre-1970 period, although it occurs within a shorter timeframe (i.e., the trend is larger). Aerosol emissions declined during the post-1970 period, hence there is also a warming effect from aerosols of 0.43±0.12 K, which is around half the greenhouse gas warming.

### 3.4 Decomposing the $F_{SW\uparrow}$ trends in the single-emissions experiments into contributions from individual cloud and aerosol variables

In this section we perform the same analysis as in Section 3.2 to quantify how much the individual changes in aerosol and cloud properties contribute to $\Delta F_{SW\uparrow}$ except for the single-emissions experiments (aerosol-only and GHG-only). It is clear from the DAMIP experiment results Figs. A1 and A3, and Table A1 (see Appendix A) that the DAMIP natural aerosol forcing, which comes mostly from volcanic aerosols, has almost no influence on the $F_{SW\uparrow}$ trends, therefore we do not consider natural aerosols further. However, there could be influences from natural aerosols that are not captured by the DAMIP natural emissions such as feedbacks between sea-spray CCN and temperature. Some of these will be represented in the experiments used here such as the effects on sea-spray from changes in wind speed as a result of temperature change. Our results (Table 1 and Fig. 6d) show that there is little change in $N_d$ in the AerChemMIP-GHG-only-proxy experiment (-1.0±0.49 cm$^{-3}$ for the pre-1970 period and 0.54±0.91 cm$^{-3}$ for the post-1970 period) in which large changes in temperature occur, which suggests little influence of climate feedbacks on CCN. Our results are likely to exclude the impact of changes in sea-spray due to changes in sea-ice coverage since we deliberately excluded sea-ice covered regions. Therefore we calculated the changes in $N_d$ for only the sea-

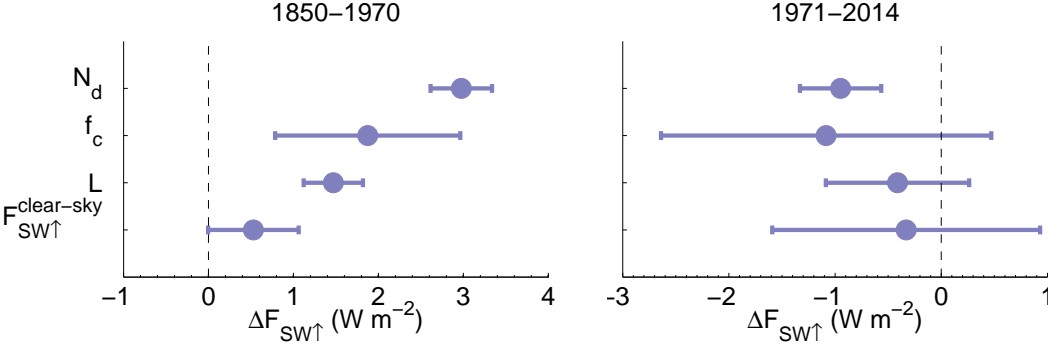

**Figure 8.** As in Fig. 5 except for the AerChemMIP aerosol only proxy.

ice regions and found values of 0.57±0.47 and -0.84±0.74cm$^{-3}$ for the pre- and post-1970 periods, respectively, suggesting
that the effect is small for this model.

### 3.4.1   Aerosol-only emissions

Figure 8 shows the contributions to $\Delta F_{\mathrm{SW\uparrow}}$ from the changes in the different aerosol and cloud variables for the AerChemMIP-
aerosol-only-proxy run calculated, as in Section 3.2, using offline radiative calculations. Percentages are quoted relative to the
offline-estimated total $\Delta F_{\mathrm{SW\uparrow}}$ for the AerChemMIP-aerosol-only-proxy (6.7±1.2 W m$^{-2}$) rather than the actual $\Delta F_{\mathrm{SW\uparrow}}$
(6.6±1.7 W m$^{-2}$). $\Delta N_{\mathrm{d}}$ provides the largest contribution during the pre-1970 period (3.0±0.36 W m$^{-2}$ or 44.2% of the total).
The $\Delta f_{\mathrm{c}}$ contribution is significantly smaller (1.9±1.1 W m$^{-2}$ or 27.8%) with the $\Delta L$ contribution (21.8%) being slightly
smaller still. The $\Delta F_{\mathrm{SW\uparrow}}^{\mathrm{clear-sky}}$ contribution is small and uncertain at 0.53±0.53 W m$^{-2}$ or 7.8%.

     The small $\Delta F_{\mathrm{SW\uparrow}}^{\mathrm{clear-sky}}$ contribution in the pre-1970 period indicates that the ARI forcing is quite small, which is consistent
with Grosvenor and Carslaw (2020). The large $\Delta N_{\mathrm{d}}$ contribution shows that the Twomey ACI effect is very important in
driving the $\Delta F_{\mathrm{SW\uparrow}}$ from aerosols. However, the contributions from changes in the cloud macrophysical properties ($f_{\mathrm{c}}$ and $L$)
are slightly more important than the Twomey ACI effect when considered together, comprising 49.1% of the $\Delta F_{\mathrm{SW\uparrow}}$ change,
compared to 44.2% from the cloud microphysical response (i.e., due to $N_{\mathrm{d}}$ changes). However, in Section 3.5.2 we show that
some of the changes in $f_{\mathrm{c}}$ and potentially in $L$ are due to cloud feedbacks that are likely to have been induced by changes in
temperature, and hence do not solely represent forcing via cloud adjustments.
For the post-1970 period the contribution to the total $\Delta F_{\mathrm{SW\uparrow}}$ (-2.7±1.9 W m$^{-2}$) from changes in $N_{\mathrm{d}}$ is -0.95±0.38 W m$^{-2}$.
The contribution from changes in $f_{\mathrm{c}}$ is also negative and of a similar magnitude, but highly uncertain (-1.1±1.6 W m$^{-2}$). The $L$
and $\Delta F_{\mathrm{SW\uparrow}}^{\mathrm{clear-sky}}$ contributions (-0.41±0.67 W m$^{-2}$ and -0.33±1.3 W m$^{-2}$ respectively) are smaller and also very uncertain.
Changes in the macrophysical cloud properties ($f_{\mathrm{c}}$ and $L$; 54.1%) therefore dominate over those of the microphysical variables
($N_{\mathrm{d}}$; 34.7%) although the macrophysical contributions are highly uncertain.

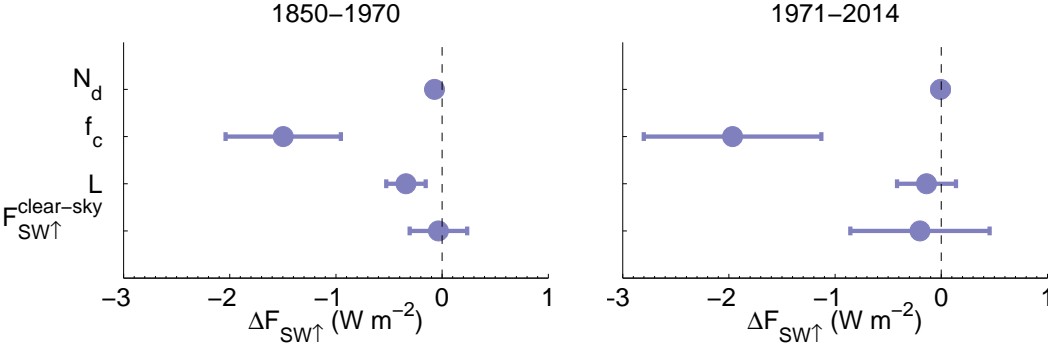

**Figure 9.** As in Fig. 5 except for the greenhouse gas only proxy.

### 3.4.2 Greenhouse gas-only emissions

The effects of greenhouse gases on $\Delta F_{SW\uparrow}$ are almost entirely driven by changes in $f_c$ for both the pre-1970 period and the post-1970 period (Fig. 9) with a larger magnitude of contribution for the post-1970 period (-2.0±0.84 vs -1.5±0.54 W m$^{-2}$ in the pre-1970 period) despite the post-1970 period being a shorter span of time. This is likely due to an enhanced rate of greenhouse gas emissions during the post-1970 period resulting in a more rapid temperature increase.

### 3.5 Aerosol forcing vs cloud-climate feedbacks

Here we examine the relative roles of aerosol forcing and feedbacks resulting from climate change (temperature, atmospheric/ocean circulation changes, etc.) on the change in $F_{SW\uparrow}$ and the cloud variables.

Aerosol forcing is the change in $F_{SW\uparrow}$ caused by a change in aerosols without a change in climate (SSTs; water vapour; atmosphere and ocean circulation, etc.). This includes rapid cloud adjustments of $f_c$ and $L$ which are potentially a major cause of changes in $F_{SW\uparrow}$.

For the greenhouse gas-only runs, we assume that the changes in $F_{SW\uparrow}$, $f_c$ and $L$ are due to climate feedbacks with no effect of greenhouse gases on cloud or clear-sky adjustments. However, we acknowledge that such effects may be possible. For example, the results of Andrews and Forster (2008) showed a -0.18 W m$^{-2}$ global change in $F_{SW\uparrow}$ from greenhouse gas adjustments (termed semi-direct forcing in that paper) for the HADGEM1 model in a doubling $CO_2$ experiment. This would represent a small fraction (6.4%) of the -2.8 W m$^{-2}$ change from the AerChemMIP-GHG-only-proxy run for the post-1970 period (although the latter is for the North Atlantic region only) and is also likely to be an overestimate for our case since the change in $CO_2$ for the post-1970 period is less than a doubling. Furthermore, Figure 7.4 of the AR6 assessment (Forster et al., 2021) estimates the global $CO_2$ adjustment effect to be around 5% of the total ERF, although this is for the combined shortwave and longwave values.

For the AerChemMIP-aerosol-only-proxy runs changes in $F_{SW\uparrow}$, $f_c$ and $L$ are split between aerosol forcing and climate feedback terms using two different methods. The first method estimates the feedback term as the change of the quantity

($\Delta X_{\text{aer}}$, where X represents either $F_{\text{SW}\uparrow}$, $f_c$ and $L$ for the AerChemMIP-aerosol-only-proxy run) minus the change in X induced by the aerosol effective radiative forcing ($\Delta X_{\text{aer}}^{\text{eff}}$):-

$$\Delta X_{\text{feedback aer}} = \Delta X_{\text{aer}} - \Delta X_{\text{aer}}^{\text{eff}} \tag{2}$$

See Box 3 in Fig. 1. Here, $\Delta X_{\text{aer}}^{\text{eff}}$ was calculated using the results from the nudged runs of Grosvenor and Carslaw (2020) (see Box 2 in Fig. 1, Section 2.7 and Appendix C).

The second method estimates the change in X due to climate feedbacks ($\Delta X_{\text{feedback aer}}^{\Delta \text{T}}$) in the AerChemMIP-aerosol-only-proxy run using the temperature change in that run ($\Delta T_{\text{aer}}$) based on the sensitivity of X to temperature in the AerChemMIP-GHG-only-proxy run :-

$$450 \quad \Delta X_{\text{feedback aer}}^{\Delta \text{T}} = \Delta T_{\text{aer}} \frac{\Delta X_{\text{GHG}}}{\Delta T_{\text{GHG}}}, \tag{3}$$

where $\Delta X_{\text{GHG}}$ and $\Delta T_{\text{GHG}}$ are the changes in X and temperature, respectively, in the AerChemMIP-GHG-only-proxy run.

The climate feedback term could include several processes. For example, aerosol and greenhouse gas forcing can change global and local temperatures and sea-ice which can then cause changes in atmospheric and oceanic circulation, and subsequent changes in the distribution of aerosols and clouds. There is evidence that warming can cause an expansion of the Hadley cell

and a poleward shift of the storm tracks (Held and Hou, 1980; Lu et al., 2007; Seidel et al., 2008) that can reduce mid-latitude cloudiness (Norris et al., 2016). Cooling would have the opposite effect, leading to increases in $F_{\text{SW}\uparrow}$ in the North Atlantic region. It has also been suggested that aerosols may have a local influence on the Atlantic Meridional Overturning Circulation (AMOC) that is more direct than the effect of aerosols on large scale temperatures (Yu and Pritchard, 2019; Robson et al., 2022). Menary et al. (2020) shows that the AMOC speeds up in the DAMIP-hist-aer run as a result of aerosol emissions and it

is feasible that changes in the AMOC could also lead to changes in cloud cover or properties and hence changes in $F_{\text{SW}\uparrow}$.

### 3.5.1   Forcing vs feedbacks for $F_{\text{SW}\uparrow}$

The balance between aerosol forcing and climate feedbacks is first examined for $\Delta F_{\text{SW}\uparrow}$. Figure 10 shows that for both periods $\Delta F_{\text{aci}}^{\text{eff}}$ is much larger than $\Delta F_{\text{ari}}^{\text{eff}}$ for the AerChemMIP-aerosol-only-proxy run. For the pre-1970 period the estimated aerosol ERF ($\Delta F_{\text{aer}}^{\text{eff}} = \Delta F_{\text{ari}}^{\text{eff}} + \Delta F_{\text{aci}}^{\text{eff}}$) of the AerChemMIP-aerosol-only-proxy run accounts for 77% of the $\Delta F_{\text{SW}\uparrow}$ of the all-emissions

run (see Table 3). Climate responses in the AerChemMIP-aerosol-only-proxy run (labelled "Aerosol Feedback" in Fig. 10) also account for 77% of the $\Delta F_{\text{SW}\uparrow}$ of the all-emissions run showing that the initial aerosol ERF and the subsequent climate feedbacks are equally important in causing changes in $F_{\text{SW}\uparrow}$ in the aerosol-only run. The $\Delta F_{\text{SW}\uparrow}$ from the AerChemMIP-GHG-only-proxy run (assumed to be all due to climate feedback) was -53% of the $\Delta F_{\text{SW}\uparrow}$ of the all-emissions run which brings the total of the aerosol forcing, aerosol-driven cloud-climate feedback and greenhouse gas-driven cloud-climate feedback terms

to 100%. Figures 6 and 7 shows that during the pre-1970 period aerosols caused a cooling of around 0.85 K in AerChemMIP-aerosol-only-proxy. This is likely to have caused a climate response that affected $F_{\text{SW}\uparrow}$ for example via an increase in cloud

fraction due to mid-latitude cloud feedbacks. The $\Delta X^{\Delta \mathrm{T}}_{\mathrm{feedback \ aer}}$ value (Eqn 3) is another estimate of this cloud-climate feedback using the above temperature change for aerosol-only emissions and is shown in Fig. 10 as the "Aerosol Feedback from $\Delta$T" datapoint. It shows good agreement with the "Aerosol Feedback" value suggesting that the local temperature change is a good indicator of the feedback contribution.

If $\Delta F_{\mathrm{SW}\uparrow}$ from the greenhouse gas-driven cloud-climate feedback (from the AerChemMIP-GHG-only-proxy run) is added to the aerosol-driven cloud-climate feedback value ("Aerosol Feedback") then we obtain an estimate of the overall change in $F_{\mathrm{SW}\uparrow}$ due to feedbacks from both types of emissions. For the pre-1970 period, this overall feedback effect on $F_{\mathrm{SW}\uparrow}$, termed "Total (Aerosol Feedback + GHG) Feedback", is considerably lower in magnitude than the aerosol forcing term and accounts for 23% of the $\Delta F_{\mathrm{SW}\uparrow}$ from the all-emissions run (cf. 77% for aerosol forcing). This indicates that in the all emissions run, which is assumed to be the run most similar to the real world, the aerosol forcing has a larger influence on $F_{\mathrm{SW}\uparrow}$ than climate feedbacks during the pre-1970 period. This dominance of aerosol forcing is mainly due to the cancellation of the warming effect of greenhouse gases and the cooling effect of aerosols (Figs. 6 and 7).

For the post-1970 period the aerosol ERF is in the opposite direction and is smaller in magnitude than for pre-1970 as expected from the smaller magnitude change in $N_{\mathrm{d}}$. The estimated change in $F_{\mathrm{SW}\uparrow}$ due to aerosol-driven cloud feedbacks is now negative in contrast to the pre-1970 period, which is consistent with the increase in temperature caused by aerosols during the post-1970 period (Figs. 6 and 7). The sign of $\Delta F_{\mathrm{SW}\uparrow}$ estimated from the temperature change ("Aerosol Feedback from $\Delta$T"; Eqn 3) is in agreement with the $\Delta F_{\mathrm{SW}\uparrow}$ due to aerosol-driven cloud feedbacks, although it is a little lower in magnitude. For the post-1970 period, the total change in $F_{\mathrm{SW}\uparrow}$ due to feedbacks associated with aerosols and greenhouse gases is considerably larger in magnitude (87% of the all-emissions run value) than the overall aerosol forcing (13%). This implies that observations of changes in $F_{\mathrm{SW}\uparrow}$ over the post-1970 period cannot be used directly to evaluate aerosol forcing in models without taking account of feedbacks.

### 3.5.2  Forcing vs feedbacks for $f_{\mathrm{c}}, N_{\mathrm{d}}, L$

The changes in cloud variables from the AerChemMIP-aerosol-only-proxy run are further split into forcing and climate feedback components in a similar way to how the the $\Delta F_{\mathrm{SW}\uparrow}$ term was split earlier, i.e., using the results from the nudged runs of Grosvenor and Carslaw (2020) (see Section 2.7 and Appendix C). Note that for $f_{\mathrm{c}}$ and $L$ it is not possible to split the forcing into ARI and ACI terms since in Grosvenor and Carslaw (2020) this could only be done for $F_{\mathrm{SW}\uparrow}$.

For the pre-1970 period (Fig. 11) slightly more of $\Delta f_{\mathrm{c}}$ (247% of the $\Delta f_{\mathrm{c}}$ of the all-emissions run) in AerChemMIP-aerosol-only-proxy comes from the climate feedback effect rather than the aerosol forcing (206%). Likewise, most of the $\Delta L$ (Fig. 12) comes from the climate feedback (93%) with 30% coming from the aerosol forcing. Hence most of the contributions to $\Delta F_{\mathrm{SW}\uparrow}$ in AerChemMIP-aerosol-only-proxy from $\Delta f_{\mathrm{c}}$ and $\Delta L$ seen in Fig. 8 are from the climate responses to the increase in aerosol rather than cloud adjustments.

For the post-1970 period the aerosol-induced changes in $f_{\mathrm{c}}$ and $L$ are negative which is consistent with the sign of the aerosol forcing. The predicted aerosol forcings are very small for both variables. The estimated climate feedback terms are larger in magnitude than the aerosol forcings, however, the uncertainties in the aerosol-induced changes are large, particularly for $L$.

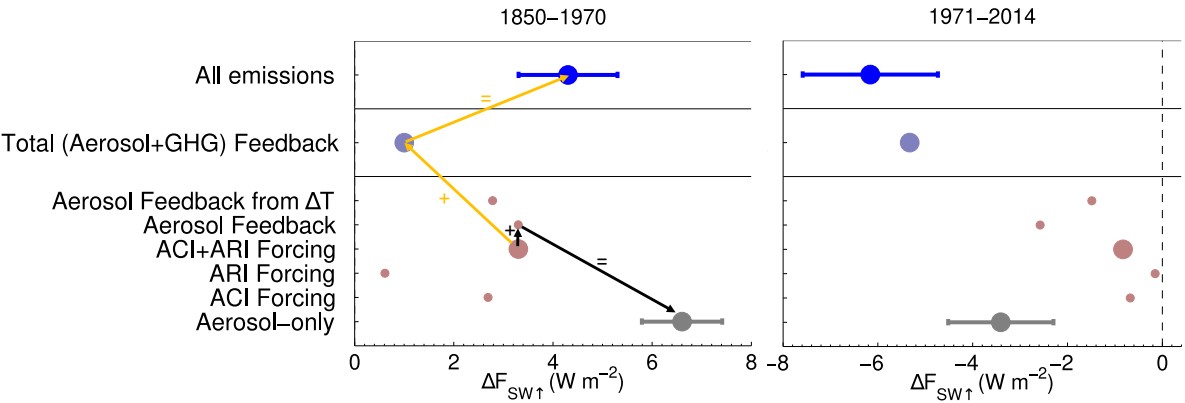

**Figure 10.** The relative roles of aerosol forcing and climate feedbacks in explaining $\Delta F_{\text{SW}\uparrow}$ between 1870 and 1970 (left column) and between 1971 and 2014 (right) for the AerChemMIP UKESM1 runs. "Aerosol-only" is the change in the AerChemMIP-aerosol-only-proxy runs as in Fig. 7 ($\Delta F_{\text{SW}\uparrow \text{ aer}}$). "ACI" and "ARI" are the aerosol effective radiative forcings ($\Delta F^{\text{eff}}_{\text{ari coupled}}$ and $\Delta F^{\text{eff}}_{\text{aci coupled}}$). "Aerosol Feedback" is the climate feedback term for the AerChemMIP UKESM1 runs ($\Delta F_{\text{SW}\uparrow \text{ feedback aer}}$) calculated using Eqn. 2 and "Aerosol Feedback from $\Delta$T" ($\Delta F^{\Delta \text{T}}_{\text{SW}\uparrow \text{ feedback aer}}$) is that calculated using Eqn. 3. "Total (Aerosol+GHG) Feedback" is the estimated total climate feedback in the all-emissions run ($\Delta F_{\text{SW}\uparrow \text{ feedback tot}}$) calculated by summing $\Delta F_{\text{SW}\uparrow}$ from the AerChemMIP-GHG-only-proxy run ($\Delta F_{\text{SW}\uparrow \text{ feedback GHG}}$) and $\Delta F_{\text{SW}\uparrow \text{ feedback aer}}$. Also shown is $\Delta F_{\text{SW}\uparrow}$ for the all-emissions UKESM1 AerChemMIP runs (AerChemMIP-all-emissions). Arrows on the left plot are drawn to indicate values that add together to give other values on the plot (see Eqn. 2 and Appendix D). These also apply to the righthand panel and to all panels for Figs. 11 and 12, but are omitted for clarity. The black arrows also apply to the DAMIP experiments (Figs A6, A7 and A8), but the orange ones do not. Arrows for $\Delta F^{\text{eff}}_{\text{aer coupled}} = \Delta F^{\text{eff}}_{\text{ari coupled}} + \Delta F^{\text{eff}}_{\text{aci coupled}}$ have also been omitted.

Fig. 7 showed that there was little change in $L$ in the AerChemMIP-GHG-only-proxy run for either period. This is a little surprising since greenhouse gas forcing caused a large reduction in $f_c$ during both periods, presumably through climate response changes. Hence, given the estimated large response of $L$ to climate responses in AerChemMIP-aerosol-only-proxy (Fig. 12), a fairly large climate response for $L$ due to greenhouse gas forcing may have been expected. It is possible that the
aerosol and greenhouse gas induced climate responses are somewhat different and have different effects on clouds, although we also note that the $L$ timeseries is particularly noisy (Fig. 6) such that the AerChemMIP-GHG-only-proxy error bar for $L$ (Fig. 7) for the post-1970 period extends into negative values and the error bar for AerChemMIP-aerosol-only-proxy in Fig. 12 is large enough to be consistent with a much smaller climate response, or even a zero climate response with the aerosol forcing accounting for all of the change. However, the uncertainties for the pre-1970 period are much smaller suggesting that the above
arguments do not apply for that period. In that case the large increase in $\Delta L$ in response to aerosol-induced climate feedbacks during the pre-1970 period when uncertainties were lower might indicate that some of the $\Delta L$ during the post-1970 period was caused by a similar circulation change in reverse (due to the opposite sign of $\Delta$T over the two periods). It is also possible that

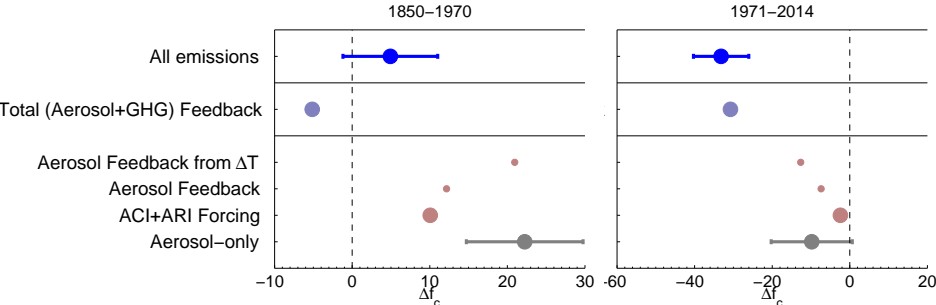

**Figure 11.** As for Fig. 10 except for $f_c$ and that the aerosol forcing term is not further split into ACI and ARI contributions.

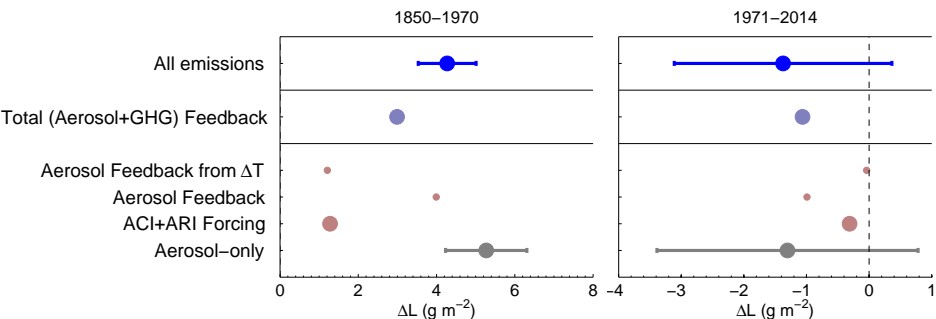

**Figure 12.** As for Fig. 10 except for $L$ and that the aerosol forcing term is not further split into ACI and ARI contributions.

the magnitude of the aerosol forcing for $L$ is underestimated, which would produce a smaller estimate for the magnitude of the climate feedback contribution for AerChemMIP-aerosol-only-proxy. Determining the reasons for the above surprising result is

left to future work.

As discussed in Section 3.3, Figs. 6 and 7 show that there is no change in either $N_d$ or $F_{SW\uparrow}^{clear-sky}$ over the two periods in the AerChemMIP-GHG-only-proxy run despite the large climate responses to greenhouse gas emissions. It is therefore likely that there was also no impact upon $N_d$ or $F_{SW\uparrow}^{clear-sky}$ from the climate responses in the AerChemMIP-aerosol-only-proxy run and hence that the changes in these variables are almost entirely driven by the aerosol changes. This suggests that almost all

of the $\Delta F_{SW\uparrow}$ that was apportioned to climate responses in the AerChemMIP-aerosol-only-proxy run (Fig. 10) was due to the associated changes in $f_c$ and $L$.

## 4    Comparison with observations

We now compare the modelled timeseries with observations. Reliable observations are only available in the later parts of the timeseries. For $F_{SW\uparrow}$ we use the Deep-C dataset (Allan et al., 2014b) that is available from 1985-2014, for $N_d$ we use MODIS

data from 2003-2012 (see Section 2.5 for details), for $\tau_a$ we use 2003-2012 Level-3 MODIS Aqua monthly mean data from

**Table 3.** Contributions to changes in $F_{SW\uparrow}$, $f_c$ and $L$ from various processes as in Figs. 10, 11, 12 along with the addition of the changes from the AerChemMIP-GHG-only-proxy run (assumed to be the climate feedback term for that run) for the UKESM1-based AerChemMIP experiments. The percentages in brackets are the contribution expressed as a percentage of the contribution of the AerChemMIP-all-emissions run.

| | | $\Delta F_{SW\uparrow}$ (W m$^{-2}$) | $\Delta f_c$ | $\Delta L$ (g m$^{-2}$) |
|---|---|---|---|---|
| Pre-1970 period | AerChemMIP-all-emissions ($\Delta X_{all}$) | 4.3±1.00 | 4.9±6.1 | 4.3±0.74 |
| | AerChemMIP-aerosol-only-proxy ($\Delta X_{aer}$) | 6.6±0.81 | 22.2±7.5 | 5.3±1.0 |
| | ACI+ARI forcing ($\Delta X^{eff}_{aer\ coupled}$) | 3.3 (77%) | 10.1 (206%) | 1.3 (30%) |
| | Feedback in AerChemMIP-aerosol-only-proxy ($\Delta X_{feedback\ aer}$) | 3.3 (77%) | 12.1 (247%) | 4.0 (93%) |
| | Feedback in AerChemMIP-GHG-only-proxy ($\Delta X_{feedback\ GHG}$) | -2.3 (-53%) | -17.3 (-353%) | -1.00 (-23%) |
| | Total (Aerosol+GHG) feedback ($\Delta X_{feedback\ tot}$) | 1.0 (23%) | -5.2 (-106%) | 3.0 (70%) |
| Post-1970 period | AerChemMIP-all-emissions ($\Delta X_{all}$) | -6.2±1.4 | -33.2±7.1 | -1.4±1.7 |
| | AerChemMIP-aerosol-only-proxy ($\Delta X_{aer}$) | -3.4±1.1 | -9.8±10.4 | -1.3±2.1 |
| | ACI+ARI forcing ($\Delta X^{eff}_{aer\ coupled}$) | -0.83 (13%) | -2.4 (7%) | -0.31 (22%) |
| | Feedback in AerChemMIP-aerosol-only-proxy ($\Delta X_{feedback\ aer}$) | -2.6 (42%) | -7.4 (22%) | -1.0 (71%) |
| | Feedback in AerChemMIP-GHG-only-proxy ($\Delta X_{feedback\ GHG}$) | -2.8 (45%) | -23.4 (70%) | -0.07 (5%) |
| | Total (Aerosol+GHG) feedback ($\Delta X_{feedback\ tot}$) | -5.4 (87%) | -30.8 (93%) | -1.1 (79%) |

the combined 550 nm Dark Target and Deep Blue product "Dark_Target_Deep_Blue_Optical_Depth_550_Combined" (Levy et al., 2013), for $f_c$ we use PATMOSx and ISCCP data from Norris et al. (2016) for 1983-2009, for $L_{all-sky}$ we use the MAC microwave satellite instrument dataset (Elsaesser et al., 2017) for 1988-2014 ($L$ is not available from this instrument), and for surface temperature we use the data from the UKESM1 atmosphere-only model run that uses observed SSTs from 1985-2014

(chosen to coincide with the $F_{SW\uparrow}$ observations).

Figure 13 shows the same timeseries as in Fig. 2, but with the observations added and with the trends shown for the period of the relevant observations. Figure 14 shows the modelled and observed trends for the two time periods along with uncertainties. It shows both the range of trends across the model ensembles and the trend from the ensemble mean (along with its uncertainty). It is clear that the modelled $F_{SW\uparrow}$ values are too high over the 1985-2014 time period and that the ensemble mean trends are

too steep. There is a reasonable amount of spread across the model ensembles, but all of the ensemble members have a stronger trend than the DEEP-C data. However, the trends from some members are within the uncertainty of the observations. The results indicate that most ensemble members have a $F_{SW\uparrow}$ trend that is too steep resulting in a $\Delta F_{SW\uparrow}$ that is too high.

Modelled $N_d$ trends and absolute values for UKESM1 are very close to those observed, although the time period is quite short and the uncertainties are large. We also note that time-mean $N_d$ in this model tends to be underestimated in the north

of the Atlantic and overestimated in the south (Grosvenor and Carslaw, 2020), hence the good agreement may disguise some compensating biases. The HadGEM model slightly underestimates the absolute values and the trend suggesting that the larger aerosol forcing seen in the UKESM1 and the larger magnitude $\Delta N_d$ values pre- and post-1970 may be more realistic.

The absolute values of $\tau_a$ match the observations (MODIS Aqua) well for UKESM1, but $\tau_a$ is overestimated by HadGEM. Since $N_d$ was slightly underestimated by HadGEM this demonstrates that $\tau_a$ is not always a good proxy for $N_d$. Similar

reasoning may explain why there is a fairly small trend in $\tau_a$ from the observations, but a fairly large trend in the observed $N_d$. The UKESM1 trend is slightly larger in magnitude than that from the observations and the trend from HadGEM is larger still. However, there is considerable uncertainty in the observed trend and considerable spread in the trends across the ensemble members such that it is difficult to conclude that the model $\tau_a$ trends are too large.

For $f_c$, the observations are not useful to evaluate the absolute magnitude since they are only provided as anomalies, but
they are useful for looking at trends. The modelled trends match the ISCCP trend well, but slightly overestimate the magnitude of the PATMOSx trend. However, the observation timeseries is very noisy and the trends are uncertain. There is also a wide spread of model trends across the ensembles showing that cloud fraction trends over these lengths of time are highly variable such that some of the ensemble members agree with both sets of observations. This makes it difficult to evaluate the model against reality; only one realisation out of a range of possibilities will have occurred in the real world. Since it was shown
earlier that changes in $f_c$ are the main driver of the changes in $F_{SW\uparrow}$ in the post 1970 period (Fig. 5), the expectation is that the model mean $f_c$ trends would be too steep in order to produce the $F_{SW\uparrow}$ trends that were too steep. This is certainly possible given the uncertainties of the observations.

The observed $L_{all-sky}$ shows no trend and a high degree of time variability, whereas the models show negative trends that look similar to the $f_c$ timeseries. Since this is the all sky liquid water path, trends will include the effect of varying $f_c$ as well
as of varying $L$. The lack of an observed trend might suggest that a small magnitude $f_c$ trend occurred in reality in order to produce the small $L_{all-sky}$ trend, or it could indicate a compensating small $L$ trend. A small $f_c$ trend would be consistent with the small observed $F_{SW\uparrow}$ trend and might indicate that the PATMOSx $f_c$ trend is more accurate so that the model $f_c$ trend magnitude is overestimated.

The surface temperatures in the model are too low, and the trends for most ensemble members and the ensemble means are
too steep. However, there is a large degree of variability across the ensemble members and some of the ensemble members do agree with the observations. The ensemble mean temperature trend being too steep is consistent with a picture of too much cloud reduction via cloud feedbacks to temperature, which would in turn cause too strong a reduction in $f_c$, $L_{all-sky}$ and $F_{SW\uparrow}$, which is consistent with the other results described in this section. It indicates that the model climate sensitivity is too strong, which may be related to the N. Atlantic cloud feedback (as also suggested in Andrews et al., 2019), but could also be due to
unrelated factors.

## 4.1 What causes the too-large $\Delta F_{SW\uparrow}$ ?

The question that arises is what causes the too-large $\Delta F_{SW\uparrow}$ in the models? Assuming that cloud feedbacks and aerosol forcing are likely the two main mechanisms that control $\Delta F_{SW\uparrow}$, we can approximate $\Delta F_{SW\uparrow}$ as :-

$$\Delta F_{SW\uparrow} = \frac{\partial F_{SW}}{\partial T} \Delta T + \Delta F_{aer}^{eff} = \lambda \Delta T + \Delta F_{aer}^{eff} \qquad (4)$$

where $T$ is the surface temperature, $\Delta F_{aer}^{eff}$ is the aerosol forcing and $\lambda$ is a measure of the cloud feedback strength. Thus, a too-large $\Delta F_{SW}$ could be due to either a cloud feedback strength ($\lambda$) that is too strong, an aerosol forcing that is too strong, or

a too-large $\Delta T$. To rule out the possibility that the $F_{\text{SW}\uparrow}$ model trend is too steep purely because of the too-large temperature trend rather than because the aerosol forcing or cloud feedback are too large we now make an estimate of the error caused by the too-large model temperature trend alone. We do this using an estimate of $\lambda$ calculated using the ratio of the change in $F_{\text{SW}\uparrow}$ over the different time periods to the change in temperature $(T)$ for the greenhouse gas only runs :-

$$\lambda = \frac{\Delta F_{\text{SW}\uparrow\text{GHG}}}{\Delta T_{\text{GHG}}} \tag{5}$$

We assume that in the greenhouse gas-only run the effect of changes in temperature on clouds via cloud feedbacks is the only factor affecting $\Delta F_{\text{SW}\uparrow}$, which is supported by Figs. 9 and A5. We then further assume that this value of $\lambda$ applies to the all-emissions runs.

Table 4 shows $\lambda$ values for different time periods. The AerChemMIP-GHG-only-proxy estimates are consistent across the different periods with values ranging between -3.4 and -3.5 W m$^{-2}K^{-1}$. The DAMIP-Hist-GHG (HadGEM-based) value for 1850-1970 (-3.4 W m$^{-2}K^{-1}$) is also consistent with these values whereas the DAMIP-Hist-GHG estimates for the 1971-1985 and 1985-2014 periods are quite different (-2.5 and -1.8 W m$^{-2}K^{-1}$). It has been noted previously that cloud feedback magnitudes can vary over time due to natural variability (Armour et al., 2013; Zhou et al., 2016; Andrews et al., 2018) and the HadGEM results may be indicative of such natural variability. Given the consistency of the UKESM1 results we therefore choose the 1985-2014 $\lambda$ value of -3.5 W m$^{-2}K^{-1}$for AerChemMIP-GHG-only-proxy since this is the period of interest when comparing with observations and noting that the HadGEM-based value was similar to this for the longer 1850-1970 period; the longer period is likely to reduce uncertainties from short-term variability. Using the larger magnitude $\lambda$ values from AerChemMIP-GHG-only-proxy also leads to an upper limit on the estimate of the temperature-bias effect (see below).

**Table 4.** $\lambda$ values ($Wm^{-2}K^{-1}$; see Equation 5) for the UKESM- (AerChemMIP-piAer) and HadGEM- (DAMIP-Hist-GHG) based greenhouse gas-only simulations (or proxies) for three different time periods.

|  | 1850-1970 | 1971-2014 | 1985-2014 |
|---|---|---|---|
| AerChemMIP-GHG-only-proxy | -3.4 | -3.4 | -3.5 |
| DAMIP-Hist-GHG (HadGEM GHG-only) | -3.4 | -2.5 | -1.8 |

Multiplying $\lambda$ by the difference between the observed and modelled $\Delta T$ values (i.e., $\Delta T_{\text{observed}} - \Delta T_{\text{model}}$) gives an estimate of the correction to the modelled $\Delta F_{\text{SW}\uparrow}$ that is needed to estimate the $\Delta F_{\text{SW}\uparrow}$ from cloud feedbacks that would be produced by using the observed temperature trend in place of the modelled one :-

$$\Delta F_{\text{SW}\uparrow,\text{corrected}} = \Delta F_{\text{SW}\uparrow} + \lambda(\Delta T_{\text{observed}} - \Delta T_{\text{model}}) \tag{6}$$

For the 1985-2014 period the corrected estimate ($\Delta F_{\text{SW}\uparrow,\text{corrected}}$) for UKESM1 is -3.6 W m$^{-2}$ (corrected from -5.0 W m$^{-2}$) and for HadGEM it is -3.1 W m$^{-2}$ (from -4.5 W m$^{-2}$). These are closer to the observed value of -1.7 W m$^{-2}$,

but are still considerably too negative. This suggests that either the model cloud feedback ($\lambda$) is too strong or the aerosol forcing is too strong. Either of these scenarios would cause a temperature increase that is too steep and hence are also consistent with these factors playing a role in causing the too-large temperature increase. We also note here that using the smaller magnitude $\lambda$ values from DAMIP-Hist-GHG would lead to a smaller correction and hence would strengthen this conclusion.

A caveat here is that it has been shown that the specific global pattern of SSTs that occurred in reality is likely to have influenced the magnitude of cloud feedbacks and the climate sensitivity (Armour et al., 2013; Zhou et al., 2016; Andrews et al., 2018) in the real world; this is known as the "pattern effect". Thus it could be the case that the model cloud feedback response (i.e., $\lambda$), and by extension the model climate sensitivity, to a given pattern and magnitude of SST changes is reasonable, but the model is not capturing the correct pattern of SSTs and hence this is why the $F_{\mathrm{SW\uparrow}}$ trend is too steep. Figures 13 and 14 also show $F_{\mathrm{SW\uparrow}}$ results from a single-member atmosphere-only (UKESM-AMIP) simulation where observed SSTs and sea-ice concentrations are imposed (see Section 2.2.3 for more details). It is clear that this run better matches the observed $F_{\mathrm{SW\uparrow}}$ timeseries and trend although the trend is still steeper than that observed. Fig. 14 shows that the trend from the atmosphere-only run is actually very similar to the estimates made in the previous paragraph where we used the observed $\Delta T$ to correct the $\Delta F_{\mathrm{SW\uparrow}}$ (converted to a trend for Fig. 14) of the all-emissions runs (UKESM1 and HadGEM). This hints that the magnitude of the SST change may be more important than the spatial pattern for $\Delta F_{\mathrm{SW\uparrow}}$ in the N. Atlantic leaving open the possibility that the cloud feedbacks or aerosol forcing in the model are incorrect. However, the uncertainties are large and further work is needed to determine this.

## 5   Discussion and conclusions

In this study we used the HadGEM global coupled climate model and the UKESM1 Earth system model to explore the factors driving historical changes in $F_{\mathrm{SW\uparrow}}$ for the North Atlantic region for ocean gridboxes that contained little sea-ice. We found that there is a positive trend in $F_{\mathrm{SW\uparrow}}$ between 1850 and 1970 and then a negative trend until 2014. The analysis shows that the pre-1970 trend is mainly driven by an increase in cloud droplet concentrations ($N_{\mathrm{d}}$) due to increases in aerosol emisons and the trend in the later period is mainly driven by a decrease in cloud fraction likely due to cloud feedbacks caused by greenhouse gas induced warming.

We also examined the relative effects of aerosol radiative forcing and climate feedbacks on the change in $F_{\mathrm{SW\uparrow}}$. In the pre-1970 period aerosol-induced cooling and greenhouse gas warming roughly counteracted each other so that there was little cloud feedback effect. Therefore, in this period aerosol forcing is the dominant cause of changes in $F_{\mathrm{SW\uparrow}}$. However, in the post-1970 period the warming from greenhouse gases intensified leading to a large warming over the North Atlantic and reduction in $F_{\mathrm{SW\uparrow}}$ from cloud feedbacks. Combined with a reduction in aerosol forcing during this period, this led to temperature feedbacks dominating over the aerosol forcing. This is summarized in the schematic of Fig. 15. These results suggests that it is unfeasible to use the post-1970 period (during which there are useful satellite observations) to evaluate and constrain ACIs, but that cloud feedbacks might be usefully evaluated. Although it may be possible to identify smaller regions or specific times during the satellite era when the aerosol effects are stronger, for example when temperature changes are small.

a)

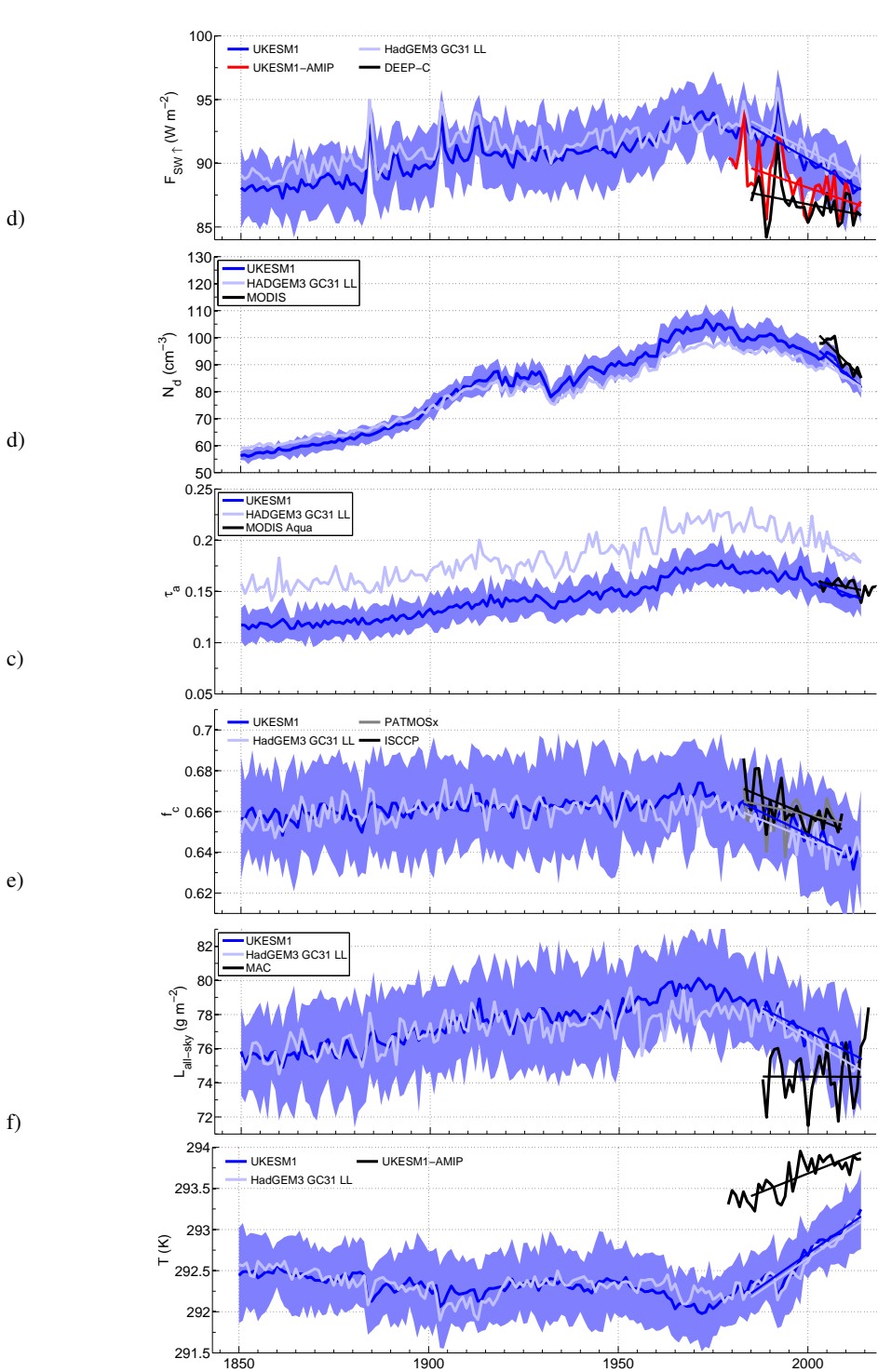

**Figure 13.** As for Fig. 2 except showing observations and trend lines that coincide with the observations for the post-1985 period. N.B. the PATMOS and ISCCP cloud fraction values are provided as anomalies from the global mean only and so the absolute values are uncertain. An arbitrary value of 0.66 was chosen to match the model values in the early part of the timeseries.

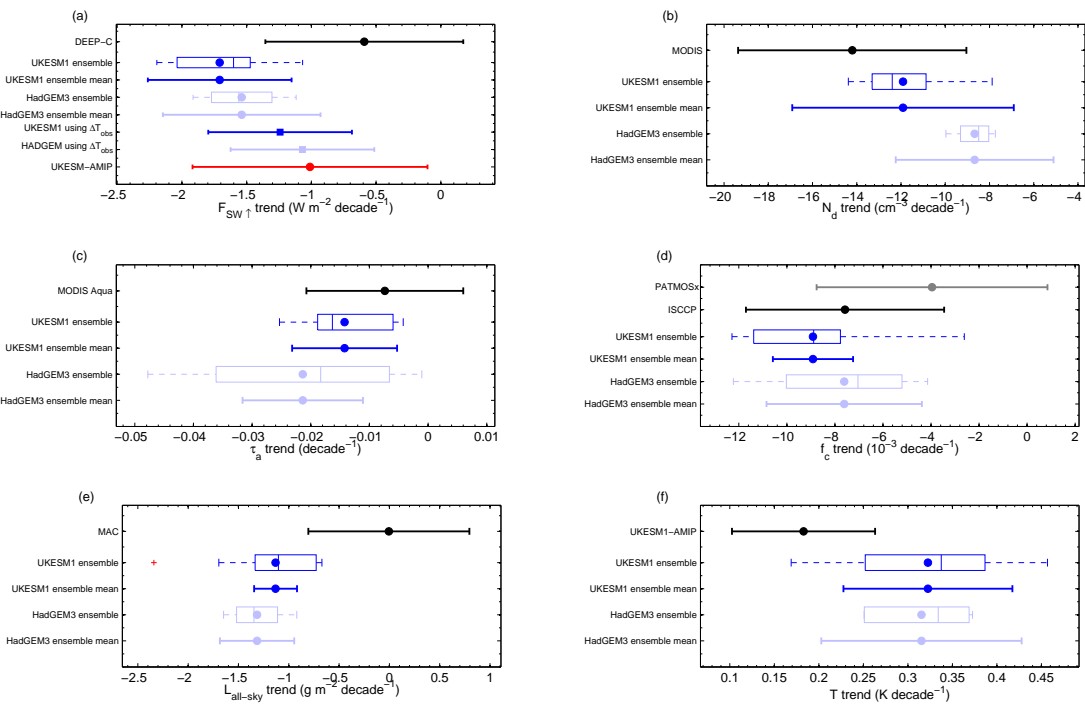

**Figure 14.** Model trends compared to observed trends for time periods chosen to match the available observations: 1985-2014 for Deep-C $F_{SW\uparrow}$ observations; 2003-2014 for MODIS $N_d$ and MODIS $\tau_a$; 1983-2009 for PATMOSx and ISCCP $f_c$; 1988-2014 for MAC LWP; 1985-2014 for the surface temperature UKESM atmosphere-only (AMIP) dataset (more data is available for surface temperature, but this period was chosen to coincide with the $F_{SW\uparrow}$ time period). For $F_{SW\uparrow}$ in (a) estimates of the model trend that would occur if the model surface temperature was correct (i.e., equal to the observed temperatures) are also shown ("using $\Delta T_{obs}$"; $\Delta F_{SW\uparrow,corrected}$ in Eqn. 6). For the models box and whisker plots of the trends across all ensemble members are shown along with the trend from the ensemble mean timeseries and its uncertainty. The box and whisker plots show the minimum and maximum as whiskers (or errors bars), except when there are outliers when the error bars are the minimum and maximum of the non-outlier values. Outliers are values that are more than 1.5 times the interquartile range away from the bottom or top of the box and are represented as plus signs. The box edges are the 25th and 75th percentiles, the line within the box is the median and the filled circle is the mean.

Comparisons to satellite observations between 1985 and 2014 indicate that the model reduction in $F_{SW\uparrow}$ is too strong for both UKESM1 and HadGEM. The simulated increase in temperature during this period is also too strong. We analysed the extent to which the too-strong temperature trend could explain the excess $\Delta F_{SW\uparrow}$ via cloud feedbacks. However, we find that the bias in temperature trend can only account for part of the $\Delta F_{SW\uparrow}$ discrepancy given the estimated model feedback strength ($\lambda = \frac{\partial F_{SW}}{\partial T}$). This suggested that UKESM1 and HadGEM have positive biases in $\lambda$ or that the negative aerosol effective radiative forcings are too strong (a too-strong aerosol forcing would produce a positive bias in the temperature increase during the 1985-2014 period because aerosol emissions declined). A $\lambda$ value that is too negative (too strong a cloud feedback) would directly impact the equilibrium climate sensitivity of the model (producing too much warming for a given forcing). Hence, biases in either the aerosol forcing or the feedback strength would have large implications for future climate projections for these models.

The analysis also hints that the "pattern effect", whereby a particular spatial pattern of SSTs has a large influence on $\lambda$ and climate sensitivity (Armour et al., 2013; Zhou et al., 2016; Andrews et al., 2018), is not having a large influence on $\lambda$ for the North Atlantic region. This conclusion is based on the result that $\Delta F_{SW\uparrow}$ from the the domain-mean timeseries for the 1985-2014 period from a simulation that used observed SSTs and sea-ice (the atmosphere-only simulation) was similar to estimates made using the UKESM1 and HadGEM coupled model data with the surface temperature changes from the domain-mean timeseries substituted for the observed temperature change; this suggests that it is the magnitude of the temperature change rather than the spatial pattern that leads to a difference in $\Delta F_{SW\uparrow}$ between the coupled and the atmosphere-only simulations for the North Atlantic. However, the result may not extend to other regions and uncertainties are large; further work is required to clarify this. Even if there was a large pattern effect, this would still require an explanation of why the model SST trends in the N. Atlantic were too steep and why the model SST pattern was incorrect. It is possible that the natural SST pattern exhibits a large degree of variability such that it might be difficult for a model to simulate the observed pattern, which may have been a low-probability event. We also note that some of the ensemble members did have reasonable N. Atlantic SST trends. On the other hand the lack of SST agreement could indicate model issues.

If the model cloud feedback strength is too large then the conclusion (based on the model results) that feedbacks are the dominant cause of the change in $F_{SW\uparrow}$ during the post-1970 period in the real world would be weakened. However, for the post-1970 period, the $\Delta F_{SW\uparrow}$ value from feedbacks would have to change from -5.4 to -0.83 W m$^{-2}$ in order for the feedback and aerosol forcing effects to be equal. Therefore, the conclusion is likely to remain robust. On the other hand, if the model aerosol forcing is too large then using the correct aerosol forcing would enhance the ratio between cloud feedback and aerosol forcing and hence strengthen the conclusions. Furthermore, the strength of the aerosol forcing was decreased during UKESM1 model development (Mulcahy et al., 2018) showing that an excessive forcing strength is a long-standing concern of the model developers.

A recent paper (Dong et al., 2022) examined the individual effects of changes in SSTs/sea-ice extent (SIE), aerosol emissions and GHG emissions for a similar region to that studied here. They used the Met Office GA6.0 atmosphere and land model (Walters et al., 2017), which is an older version of the climate model used in this study. They used atmosphere-only simulations with SSTs taken from observations and examined differences between 2000-2015 time averages and 1980-1985 time averages,

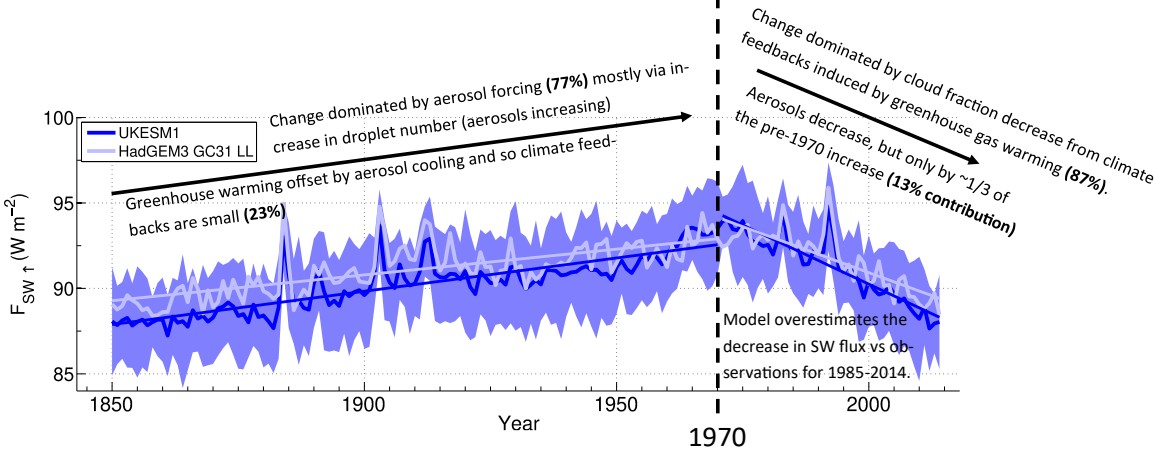

**Figure 15.** Schematic showing the main influences on the determination of the change in $F_{SW\uparrow}$ during the pre-1970 and post-1970 periods. The quoted percentages are the percentage contributions to $\Delta F_{SW\uparrow}$ from aerosol forcing and climate feedbacks for the two periods.

which is within the post-1970 analysed in our paper. They focused on the June, July, August (JJA) period. Their results showed
that aerosol emission changes dominated the change in downwelling surface solar radiation (SSR) with little influence from
SSTs/SIE or GHGs. The lack of influence from SSTs/SIE for that period is in contrast to our results where the cloud feedbacks
(driven by SST changes) dominated over aerosol forcing in terms of producing changes in $F_{SW\uparrow}$ (and presumably SSR too).
This difference in results could be due to a number of reasons. One is their use of observed SSTs in contrast to our simulations
where the SSTs were predicted by the coupled ocean model. We showed earlier that the SST trends in our coupled model
simulations for the post-1970 period were too strong, although correcting for that bias did not change our conclusions. Another
potential reason is that there have been a number of advancements of the model between the version used in Dong et al. (2022)
and that used in our paper. Those changes are likely to have affected the model feedback responses as well as the aerosol
responses and hence a different balance of aerosol forcing to feedbacks is perhaps expected. Finally, they focused on the JJA
season, whereas we used annual averages. Further work is recommended to determine the reasons for these differences as well
as to examine differences amongst various other models.

A final interesting implication that follows from our results is that the appearance of coincident peaks in the $N_d$ and $F_{SW\uparrow}$
timeseries from the UKESM1 and HadGEM models at around 1970 is due to chance. The decrease in $F_{SW\uparrow}$ after 1970 is almost
entirely caused by the growing effects of greenhouse gas emissions on the larger scale atmospheric and/or ocean circulation
rather than the decrease in aerosols that also starts around 1970. Hence if the greenhouse gas related effects were shifted to
earlier or later in the timeseries (e.g., due to the rapid increase in greenhouse gas emissions occuring earlier or later) we would
expect the decline in $F_{SW\uparrow}$ to occur correspondingly earlier or later such that the peaks would no longer be coincident. This
can be contrasted to the situation over land where the turning point in surface SW flux has been associated with a decline in
aerosol emissions (Wild et al., 2005).

## Appendix A: HadGEM DAMIP results

### A1    North Atlantic timeseries for HadGEM

Fig. A1 shows the timeseries of $F_{SW\uparrow}$ and the cloud variables expressed as an anomaly relative to the 1850-1859 mean for the DAMIP experiments. This is similar to the AerChemMIP proxy timeseries shown in Fig. 6 except that for the DAMIP experiments the different emission types (aerosol, greenhouse gases and natural emissions) were applied individually and so there is no need to approximate the effects of greenhouse gas-only and aerosol-only emissions as for AerChemMIP (see Section 2.2.2). Aerosol emissions (DAMIP-Hist-Aer) cause an increase in $F_{SW\uparrow}$ over time whereas greenhouse gas emissions (DAMIP-Hist-GHG) cause a decrease. Natural aerosols (DAMIP-Hist-Nat) produce no trend over the historical period. The sum of the $F_{SW\uparrow}$ perturbations from the single emissions runs matches the total from the HadGEM run (with all emissions) reasonably well suggesting that the main forcing agents are accounted for and that the effects of the individual forcing agents can be combined in a linear sum to approximate the overall change in the full model.

### A2    Decomposing the $F_{SW\uparrow}$ trends in HadGEM into contributions from individual variables

The results of the offline radiative calculations to quantify the effect of changes in cloud variables to the change in $F_{SW\uparrow}$ for the HadGEM model (Figure A2) are very simillar to those from UKESM1 with $N_d$ changes contributing the most to $\Delta F_{SW\uparrow}$ for the pre-1970 period and $f_c$ changes contributing the most for the post-1970 period.

### A3    Quantifying the effects of individual emission types on $F_{SW\uparrow}$ and cloud variable changes

#### A3.1    Effect of emissions on $F_{SW\uparrow}$

Figure A3 summarizes the contributions of each emission type to $\Delta F_{SW\uparrow}$ in the two periods for the HadGEM model. For the pre-1970 period the $\Delta F_{SW\uparrow}$ due to aerosol emissions (estimated from the DAMIP-hist-aer simulation) is 5.3±0.81 W m$^{-2}$ (see Table A1 for the values), which is much larger in magnitude than the reduction in $F_{SW\uparrow}$ caused by greenhouse gas emissions (from DAMIP-Hist-GHG; -1.9±0.46 W m$^{-2}$). However, this reduction is still important and shows that in the models with all emissions applied the effect of SW aerosol forcing is offset by around 36% by opposing greenhouse gas effects.

For the post-1970 period there is very little contribution from aerosol emissions (0.10±1.1 W m$^{-2}$) despite a reduction in $N_d$ that is 40% of the magnitude of the increase of the pre-1970 period. This differs from the results from the AerChemMIP proxy (Fig. 7) where the negative contribution to $\Delta F_{SW\uparrow}$ from the reduction in aerosol emissions during the post-1970 period was estimated to be larger in magnitude than that from the greenhouse gases emissions increase. For the DAMIP experiment, there is a relatively large negative contribution from greenhouse gas emissions (-2.3±0.68 W m$^{-2}$) for the post-1970 period. For both periods there is very little contribution to $\Delta F_{SW\uparrow}$ from natural emissions, which justifies the assumption that aerosols and greenhouse gases are main drivers of changes in $F_{SW\uparrow}$ that was made for the AerChemMIP calculations.

### A3.2 Effect of emissions on $f_c$, $N_d$ and $L$

Here we consider how the individual emission types affect the underlying cloud variables that were shown in the previous sections to drive the $F_{SW\uparrow}$ changes (Fig. A3).

The pre-1970 results for $f_c$ for the DAMIP experiments are very similar to those from AerChemMIP with opposing effects on $f_c$ from greenhouse gases and aerosols to give little overall $f_c$ change. However, post-1970 aerosol emissions actually cause a small increase in $f_c$ for DAMIP, which is consistent with the near-zero change in $F_{SW\uparrow}$ from aerosols. In contrast, for AerChemMIP aerosols caused a decrease in $F_{SW\uparrow}$ and $f_c$.

The results for changes in $L$ are broadly similar between the DAMIP and AerChemMIP, particularly for the pre-1970 period. However, for the post-1970 peroid the error bars are quite large, which likely explains any differences.

### A3.3 Effect of emissions on surface temperature

For the pre-1970 period the DAMIP results are simlar to the AerChemMIP ones. For the post-1970 period aerosol emissions in the DAMIP experiment cause the surface temperature to decrease slightly, whereas in AerChemMIP they caused a relatively large increase. This opposing behaviour is consistent with the decrease in $F_{SW\uparrow}$ caused by aerosols in AerChemMIP and near-zero change in $F_{SW\uparrow}$ in DAMIP since it is likely that the decrease in $F_{SW\uparrow}$ in AerChemMIP might cause a warming, but also the warming was shown to causes a decrease in $F_{SW\uparrow}$ through cloud feedbacks (and vice versa for DAMIP).

### A4 Decomposing the $F_{SW\uparrow}$ trends in the single-emissions experiments into contributions from individual cloud and aerosol variables

### A4.1 Aerosol-only emissions

The results for DAMIP-hist-aer (Fig. A4) are very similar to those for the AerChemMIP experiments for both periods. Changes in $N_d$ drive the majority of the change in $F_{SW\uparrow}$ for both periods. Changes in $f_c$ and $L$ are of lesser importance and only for the pre-1970 period.

### A4.2 Greenhouse gas-only emissions

The results for the DAMIP-hist-GHG simulations (Fig. A5) are very similar to those from AerChemMIP for both periods.

### A5 Aerosol forcing vs cloud-climate feedbacks

### A5.1 Forcing vs feedbacks for $F_{SW\uparrow}$

The DAMIP results (Fig. A6) are similar to those from AerChemMIP for the pre-1970 period. However, for the post-1970 period cloud-climate feedbacks in the aerosol-only DAMIP-hist-aer simulation drive an increase in $F_{SW\uparrow}$ (consistent with the decrease in surface temperature) as opposed to a decrease in $F_{SW\uparrow}$ in AerChemMIP (consistent with the increase in surface temperature). As a result the overall cloud-climate feedback term driven by the combination of aerosols and greenhouse gases

is smaller for DAMIP than for AerChemMIP. However, the overall feedback is still larger than the aerosol radiative forcing for DAMIP/HadGEM.

## A5.2 Forcing vs feedbacks for $f_c$, $N_d$, $L$

For the pre-1970 period the DAMIP results for $f_c$ (Fig. A7) are again similar to those from AerChemMIP. For the post-1970 period the cloud feedbacks cause an increase in $f_c$ for DAMIP-hist-aer, whereas they caused a decrease in AerChemMIP, which is consistent with the respective changes in temperature.

For the pre-1970 period the DAMIP results for changes in $L$ (Fig. A8) are similar to those from AerChemMIP. As for AerChemMIP there is a large estimated change in $L$ due to the cloud-climate feeedback term in the aerosol-only run when calculated as the difference between the total change in $L$ and the estimated change from aerosol radiative forcing. Again, though, this feedback term is a lot larger than the feedback term estimated using $\lambda$ (Eqn. 5) from the greenhouse-gas only simulation and $\Delta T$ from the aerosol-only simulation. This suggests that this discrepancy is not due to the particular model setup of either UKESM1 or HadGEM and that it is robust result between the two sets of ensemble runs. We can speculate that the discrepancy might be due to the temperature change in the NA not being the controlling factor for cloud feedbacks onto $L$ (e.g., the temperature change elsewhere may be more important), or that this result is spurious due to the noisy nature of the $L$ timeseries (Fig. A1). Further research is needed to fully determine the cause.

For the post-1970 period the overall feedback term for $L$ for HadGEM is positive, whereas it was negative for UKESM1. This is partly driven by a positive feedback term in DAMIP-hist-aer (instead of negative in AerChemMIP-aerosol-only-proxy) and a larger feedback term in DAMIP-hist-GHG than in AerChemMIP-GHG-only-proxy. Again, though the timeseries are noisy and the error bars in Figs. 12 and A7 are large so that confidence in this result is low.

## A6 Summary

The HadGEM-based DAMIP results are broadly similar to those from the UKESM1-based AerChemMIP experiments. The most prominent discrepancy is the lack of reduction in $F_{SW\uparrow}$, $f_c$ and surface temperature during the post-1970 period for the DAMIP-hist-aer (aerosol-only) ensemble; all of these quantities reduce for the AerChemMIP-aerosol-only-proxy ensemble. Here we can only speculate about possible reasons for this; further work would be needed to draw conclusions. One possibility is that the dedicated aerosol-only DAMIP-hist-aer simulation allows the AMOC to increase in strength until 1970 due to the increase in aerosols over that period, which is a proven effect of aerosol forcing in many models (Menary et al., 2020; Robson et al., 2022). This may prevent a rapid response of the climate in the NA to the post-1970 reduction in aerosols due to inertia in the AMOC perhaps related to ocean heat storage, sea-ice changes, etc. Such effects would not be captured by the estimate of aerosol-only effects from the AerChemMIP-aerosol-only-proxy timeseries. Another possibility is that the temperature and cloud-climate feedbacks in DAMIP-hist-aer are being controlled by changes in aerosols outside of the NA region where aerosols may continue to rise after 1970 (e.g., Asia). This idea is supported by the dominance of cloud feedbacks in determining the change in $f_c$ in DAMIP-hist-aer for the post-1970 period (Fig. A7). This hypothesis would require an explanation for

 why this result is not seen in the AerChemMIP-aerosol-only-proxy ensemble, suggesting a non-linearity between the effects of the aerosol-only and greenhouse gas-only simulations in explaining the all-emissions simulation results.

**Table A1.** As for Table 1 except for the HadGEM model and DAMIP experiment runs.

| Period | | $\Delta F_{\mathrm{SW\uparrow}}$ $(Wm^{-2})$ | $\Delta F_{\mathrm{SW\uparrow}}^{\mathrm{clear-sky}}$ $(Wm^{-2})$ | $\Delta f_c$ (x $10^{-3}$) | $\Delta N_d$ $(cm^{-3})$ | $\Delta \tau_a$ (x $10^{-2}$) | $\Delta L$ $(gm^{-2})$ | $\Delta T$ $(K)$ |
|---|---|---|---|---|---|---|---|---|
| 1850- | HadGEM | 3.6±0.99 | 1.7±0.87 | 6.7±4.6 | 35.0±7.4 | 5.6 ± 0.9 | 2.3±0.65 | -0.19±0.23 |
| 1970 | DAMIP hist-aer | 5.3±0.81 | 1.8±0.25 | 18.9±5.6 | 35.0±6.5 | 5.3 ± 0.7 | 3.1±0.64 | -0.81±0.20 |
| | DAMIP hist-GHG | -1.9±0.46 | -0.05±0.09 | -16.0±4.3 | 0.32±0.34 | 0.25 ± 0.41 | -0.03±0.57 | 0.58±0.09 |
| | DAMIP hist-nat | 0.45±0.76 | -0.04±0.81 | 3.3±4.5 | 0.14±0.39 | 0.30 ± 0.54 | -0.02±0.71 | -0.06±0.10 |
| | DAMIP sum | 3.9±1.5 | 1.8±0.82 | 6.2±10.8 | 35.5±5.1 | 5.8 ± 0.8 | 3.0±1.2 | -0.30±0.25 |
| | | | | | | | | |
| 1971- | HadGEM | -4.6±1.8 | -1.9±1.8 | -26.5±6.0 | -14.4±4.0 | -4.2 ± 0.9 | -0.49±1.2 | 0.96±0.38 |
| 2014 | DAMIP hist-aer | 0.10±1.1 | -1.1±0.35 | 5.9±6.6 | -15.1±3.2 | -4.0 ± 1.4 | -0.04±0.95 | -0.25±0.21 |
| | DAMIP hist-GHG | -2.3±0.68 | -0.15±0.18 | -22.8±6.5 | 0.16±0.63 | 0.12 ± 0.74 | 0.84±1.2 | 0.90±0.13 |
| | DAMIP hist-nat | 0.03±1.1 | -0.63±1.6 | 1.9±6.6 | -0.03±0.59 | -0.38 ± 0.75 | 0.91±1.4 | -0.02±0.16 |
| | DAMIP sum | -2.1±2.0 | -1.9±1.7 | -14.9±10.6 | -15.0±3.1 | -4.3 ± 1.8 | 1.7±2.1 | 0.63±0.23 |

**Table A2.** As for Table 2 except for the DAMIP experiments.

| | | HadGEM | | Hist-Aer | | Hist-GHG | |
|---|---|---|---|---|---|---|---|
| | | 1850-1970 | 1971-2014 | 1850-1970 | 1971-2014 | 1850-1970 | 1971-2014 |
| Actual $\Delta F_{\mathrm{SW\uparrow}}$ (W m$^{-2}$) | | 3.6±0.99 | -4.6±1.8 | 5.3±0.81 | 0.10±1.1 | -1.9±0.46 | -2.3±0.68 |
| Offline $\Delta F_{\mathrm{SW\uparrow}}$ (W m$^{-2}$) | | 4.5±0.71 | -4.1±1.1 | 5.0±0.63 | -0.52±0.64 | -1.3±0.34 | -1.9±0.49 |
| Contribution to $\Delta F_{\mathrm{SW\uparrow}}$ (W m$^{-2}$) from:- | | | | | | | |
| $f_c$ | | 0.58±0.25 (12.9%) | -2.4±0.59 (58.1%) | 1.5±0.50 (30.7%) | 0.49±0.58 (-93.8%) | -1.3±0.37 (98.3%) | -2.0±0.61 (105.2%) |
| $N_d$ | | 3.0±0.36 (66.0%) | -0.99±0.31 (24.0%) | 2.2±0.35 (44.2%) | -0.77±0.16 (147.6%) | 0.05±0.03 (-3.7%) | -0.02±0.05 (0.84%) |
| $L$ | | 0.55±0.17 (12.1%) | -0.21±0.28 (5.2%) | 0.81±0.17 (16.3%) | 0.05±0.22 (-9.0%) | -0.03±0.14 (2.0%) | 0.09±0.27 (-4.9%) |
| Clear-sky $F_{\mathrm{SW\uparrow}}$ | | 0.51±0.27 (11.3%) | -0.56±0.63 (13.6%) | 0.53±0.07 (10.7%) | -0.30±0.16 (58.5%) | 0.03±0.04 (-2.5%) | -0.07±0.09 (3.9%) |
| Surface albedo | | 0.03±0.03 (0.74%) | -0.01±0.08 (0.31%) | 0.03±0.03 (0.70%) | -0.01±0.08 (1.7%) | 0.03±0.03 (-2.6%) | -0.02±0.07 (0.81%) |
| Residual | | -0.14 (-3.0%) | 0.05 (7.1%) | -0.13 (-2.6%) | 0.03 (62.4%) | -0.11 (8.4%) | 0.11 (2.9%) |

## Appendix B:  Testing the assumptions made for AerChemMIP aerosol and greenhouse gas-only proxies

Here we examine the DAMIP single-emission experiment results. The DAMIP experiments may provide some extra insight into the range of possible behaviour given the likely large degree of natural variability. Furthermore, they are based on the HadGEM model rather than the UKESM1 model and hence may display some different behaviour due the slightly different model physics and settings. We also use the DAMIP experiments in order to validate some of the assumptions made when using the AerChemMIP experiment to approximate single-emission experiments. For example, for the AerChemMIP experiment there

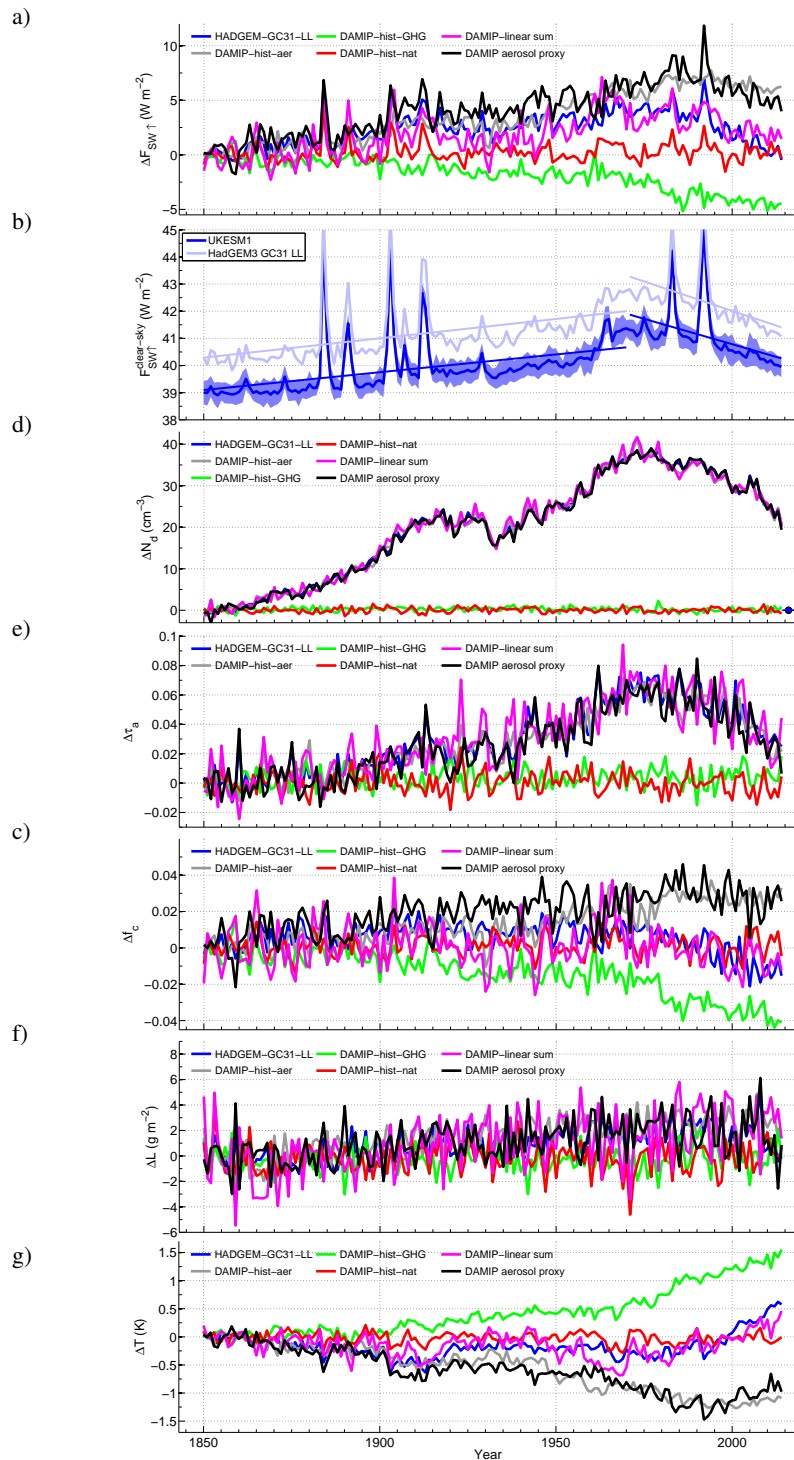

**Figure A1.** As for Fig. 6 except for the various single-forcing DAMIP simulations. Lines are shown for the HadGEM-GC3.1 model (which is the model used for the DAMIP experiments) with all forcings applied, the aerosol only forcing DAMIP run (DAMIP-hist-aer), the greenhouse gas only forcing run (DAMIP-hist-GHG), the natural only forcing run (DAMIP-hist-nat), the sum of the perturbations from all of the single forcing runs (DAMIP-linear-sum) and the estimate of the aerosol-only emissions perturbations calculated by subtracting the greenhouse gas-only values from the all-emissions HadGEM values (DAMIP aerosol proxy).

**Table A3.** As for Table 3 except for the HadGEM-based DAMIP experiments.

| | | $\Delta F_{SW\uparrow}$ (W m$^{-2}$) | $\Delta f_c$ | $\Delta L$ (g m$^{-2}$) |
|---|---|---|---|---|
| Pre-1970 period | All emissions (HadGEM ensemble; $\Delta X_{all}$) | 3.6±0.99 | 6.7±4.6 | 2.3±0.65 |
| | DAMIP Hist-Aer aerosol-only ($\Delta X_{aer}$) | 5.3±0.81 | 18.9±5.6 | 3.1±0.64 |
| | ACI+ARI forcing ($\Delta X_{aer\ coupled}^{eff}$) | 2.4 (67%) | 7.2 (107%) | 0.93 (40.4%) |
| | Climate feedback in DAMIP-HistAer ($\Delta X_{feedback\ aer}$) | 2.9 (81%) | 11.6 (173%) | 2.17 (94.3%) |
| | Climate feedback in DAMIP-HistGHG ($\Delta X_{feedback\ GHG}$) | -1.9 (-53%) | -16.0 (-239%) | -0.03 (-1%) |
| | Total (Aerosol+GHG) feedback ($\Delta X_{feedback\ tot}$) | 1.0 (27.8%) | -4.4 (-65.7%) | 2.1 (91.3%) |
| Post-1970 period | All emissions (HadGEM ensemble; $\Delta X_{all}$) | -4.6±1.8 | -26.5±6.0 | -0.49±1.2 |
| | DAMIP Hist-Aer aerosol-only ($\Delta X_{aer}$) | 0.10±1.1 | 5.9±6.6 | -0.04±0.95 |
| | ACI+ARI forcing ($\Delta X_{aer\ coupled}^{eff}$) | -0.63 (14%) | -1.8 (7%) | -0.24 (49.0%) |
| | Climate feedback in DAMIP-HistAer ($\Delta X_{feedback\ aer}$) | 0.8 (-17%) | 7.8 (-29%) | 0.21 (-42.9%) |
| | Climate feedback in DAMIP-HistGHG ($\Delta X_{feedback\ GHG}$) | -2.3 (50%) | -22.8 (86%) | 0.84 (-171%) |
| | Total (Aerosol+GHG) feedback ($\Delta X_{feedback\ tot}$) | -1.5 (32.6%) | -15.0 (56.6%) | 1.1 (-224.5%) |

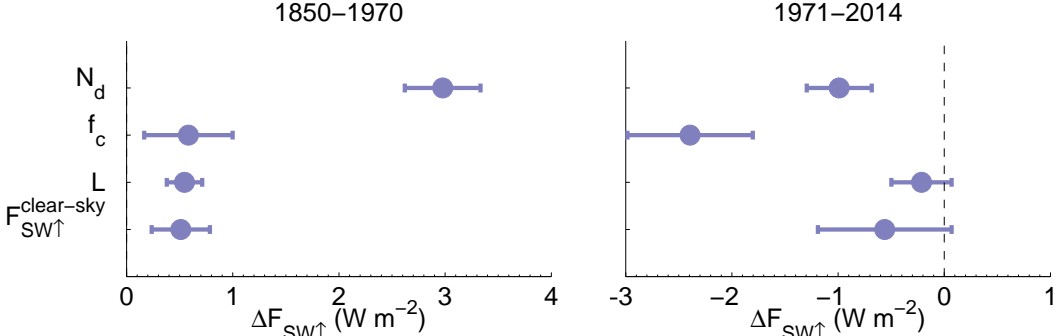

**Figure A2.** As for Fig. 5 except for the HadGEM model.

is not a true aerosol-only or greenhouse gas-only simulation as there are for the DAMIP experiment. We make the assumption that aerosols and greenhouse gases are the main factors that cause changes in the variables of interest. This means that the AerChemMIP-piAer experiment, where the full set of historical emissions are used except for aerosols for which pre-industrial (PI) emissions are used, would be equivalent to a greenhouse gas-only experiment. A proxy for an aerosol emissions-only experiment was estimated by subtracting the timeseries from AerChemMIP-piAer from the full emissions simulations. We do the same here for the DAMIP experiment to estimate the accuracy of the AerChemMIP estimate.

Fig A1 shows the proxy aerosol emissions-only timeseries calculated using the DAMIP results (HadGEM minus DAMIP-hist-GHG). Comparison of these with the DAMIP-hist-aer (true aerosol-only emission experiment) timeseries shows that the two timeseries and trends are very similar for all variables suggesting that the approach used for the calculation of the AerChemMIP-aerosol-only-proxy timeseries is valid. Table B1 lists those results for $F_{SW\uparrow}$. The estimated change in $F_{SW\uparrow}$ (aerosol proxy) for the pre-1970 period is very similar to that from the DAMIP-hist-aer experiment suggesting that the proxy

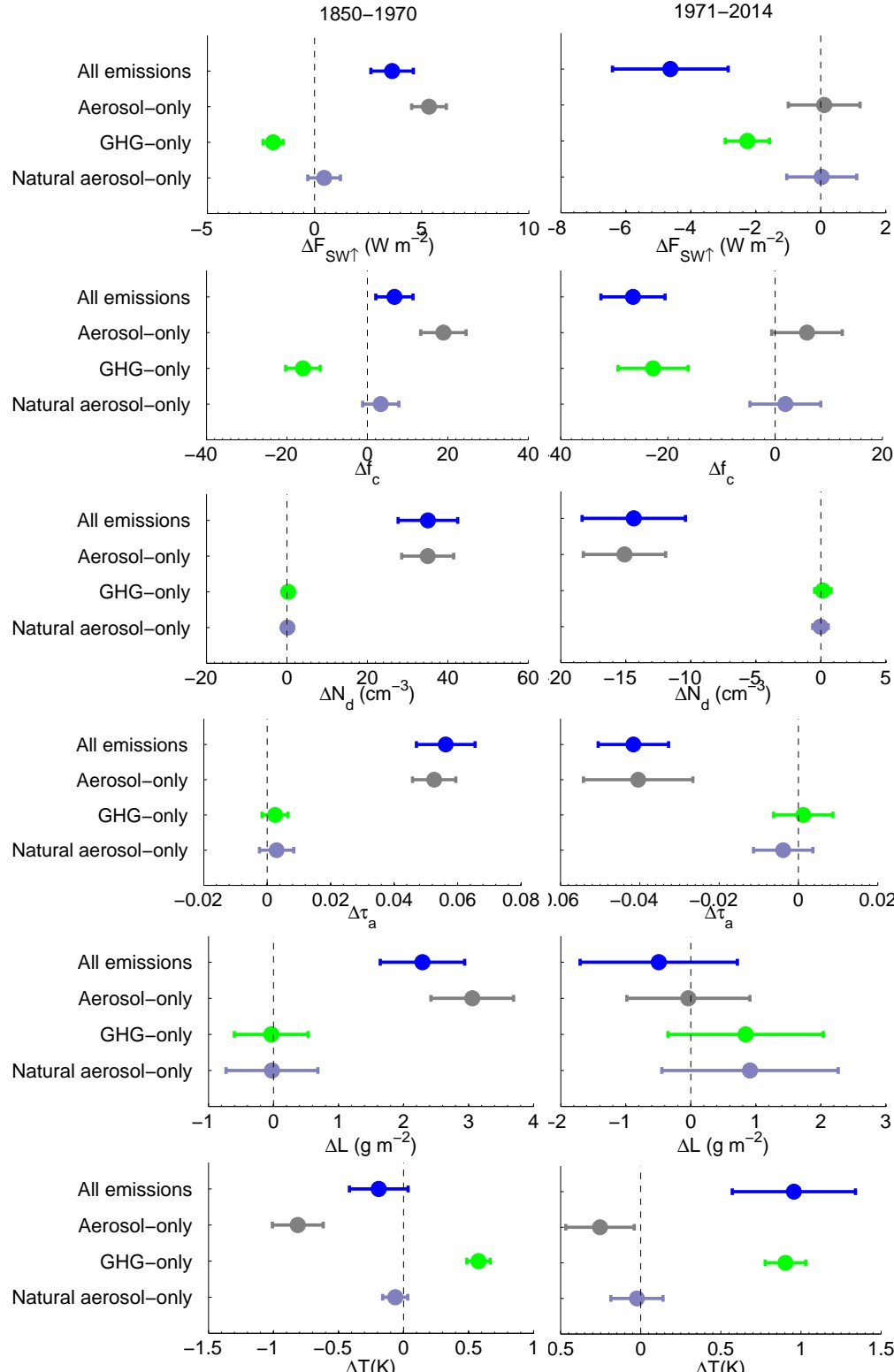

**Figure A3.** As for Fig. 7 except for the DAMIP and HadGEM simulations. A natural aerosol-only bar is now shown from the DAMIP-HistNat experiment.

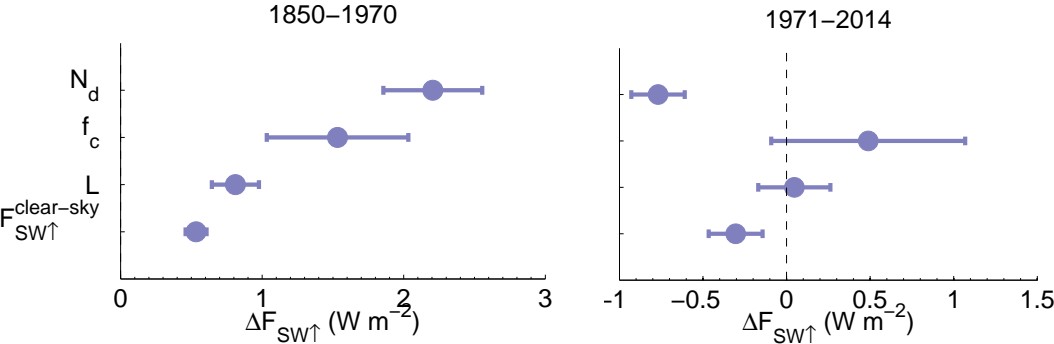

**Figure A4.** As for Fig. 8 except for the aerosol-only emissions run (DAMIP-HistAer).

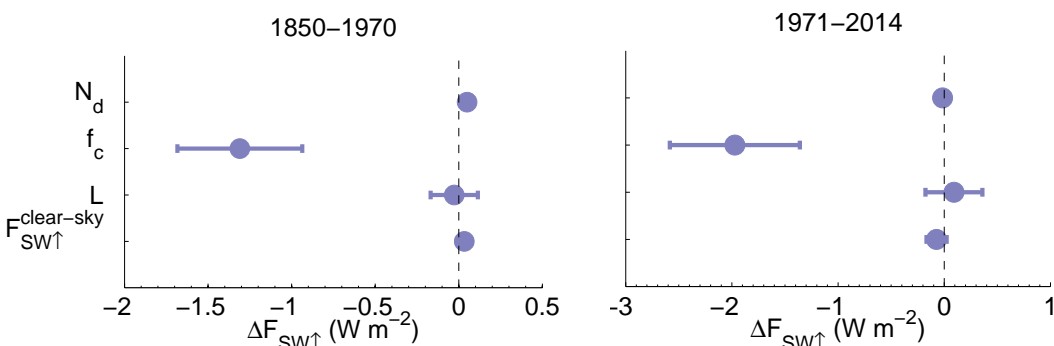

**Figure A5.** As for Fig. A5 except for the greenhouse gas-only emissions run (DAMIP-HistGHG).

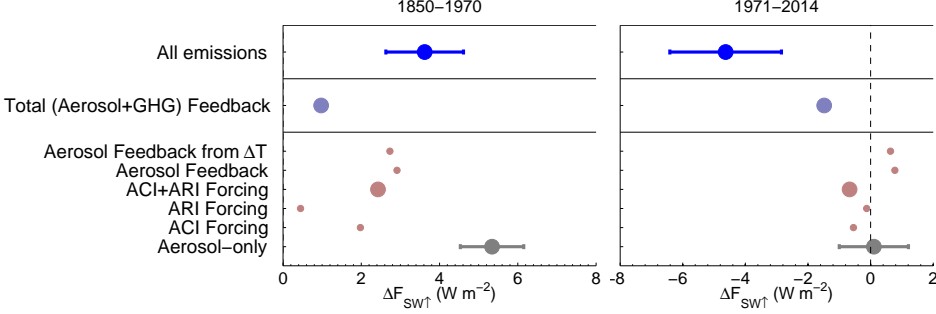

**Figure A6.** As for Fig. 10 except for the HADGEM-based DAMIP experiments.

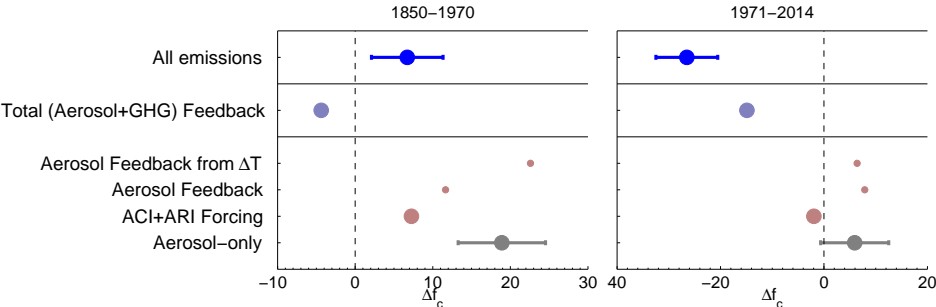

**Figure A7.** As for Fig. 11 except for the HADGEM-based DAMIP experiments.

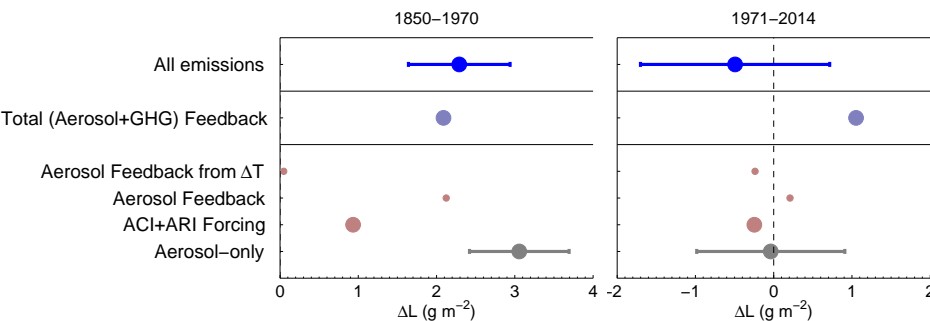

**Figure A8.** As for Fig. 12 except for the HADGEM-based DAMIP experiments.

works well. For the post-1970 period the uncertainties in the $\Delta F_{\mathrm{SW\uparrow}}$ values are large for DAMIP-hist-aer and the proxy so that

a comparison is not meaningful.

The table also compares the results of the offline radiative calculations used to estimate the change in $F_{\mathrm{SW\uparrow}}$ and contributions to the changes in $F_{\mathrm{SW\uparrow}}$ from changes in the different cloud variables for the aerosol-only runs. For the pre-1970 period the aerosol proxy values agree with the DAMIP-hist-aer values within the uncertainties suggesting that the aerosol proxy estimates are sufficient for the AerChemMIP UKESM1 analysis in the main text. The uncertainties are very large for the post-1970 period

and so it is difficult to assess the accuracy of the aerosol proxy method. However, the offline analysis of the contributions to the change in $F_{\mathrm{SW\uparrow}}$ from the change in cloud variables is highlighted as being uncertain in the main text and is not relied upon for the conclusions that are drawn. We also note that the use of the aerosol proxy only applies for the analysis of the aerosol-only emissions and not the other runs.

## Appendix C: Estimation of aerosol forcing based on nudged simulations

Here we utilize output from the same controlled meteorology (nudged) simulations as used in Grosvenor and Carslaw (2020), but for the region of the North Atlantic used in the current paper, to estimate the changes due to aerosol that would occur

**Table B1.** As for Table 2 except for the DAMIP experiments for which a comparison is made of the changes in $F_{SW\uparrow}$ and the offline radiative calculations for the aersol-only emissions runs (DAMIP-hist-aer) and the estimate of aerosol-only emissions (Aerosol Proxy).

| | | 1850-1970 | | 1971-2014 | |
| | | Hist-Aer | Aerosol Proxy | Hist-Aer | Aerosol Proxy |
| --- | --- | --- | --- | --- | --- |
| Actual $\Delta F_{SW\uparrow}$ (W m$^{-2}$) | | 5.3±0.81 | 5.50±1.45 | 0.10±1.1 | -2.30±2.48 |
| Offline $\Delta F_{SW\uparrow}$ (W m$^{-2}$) | | 5.0±0.63 | 5.80±1.05 | -0.52±0.64 | -2.20±1.59 |
| Contribution to $\Delta F_{SW\uparrow}$ (W m$^{-2}$) from:- | | | | | |
| $f_c$ | | 1.5±0.50 (30%) | 1.88±0.79 (32%) | 0.49±0.58 (-94%) | -0.40±1.20 (18%) |
| $N_d$ | | 2.2±0.35 (44%) | 2.95±0.39 (51%) | -0.77±0.16 (148%) | -0.97±0.36 (44%) |
| $L$ | | 0.81±0.17 (16%) | 0.58±0.31 (10%) | 0.05±0.22 (16%) | -0.30±0.55 (10%) |
| Clear-sky $F_{SW\uparrow}$ | | 0.53±0.07 (11%) | 0.48±0.31 (8%) | -0.30±0.16 (11%) | -0.49±0.72 (8%) |
| Surface albedo | | 0.03±0.03 (1%) | 0.00±0.06 (0%) | -0.01±0.08 (1%) | 0.01±0.15 (0%) |

in the coupled simulations (UKESM1, HadGEM, AerChemMIP and DAMIP) if there were no responses of the large scale atmospheric or ocean circulation to the aerosol forcing, as was the case in the Grosvenor and Carslaw (2020) simulations where the nudging and use of prescribed SSTs prevents such responses. From the Grosvenor and Carslaw (2020) simulations

changes in various quantities and a change in $N_d$ were calculated, and therefore the sensitivities of the quantities to $N_d$ changes that result from changing from 1850 to 2009 aerosol emissions could also be calculated. We assume that the same sensitivity (when expressed in relative form) would occur in the coupled runs and then scale these sensitivities by the $\Delta N_d$ from the coupled runs for both the pre-1970 period and the post-1970 period to estimate the change in quantity $x$, where $x$ is either $\Delta F_{ari}^{eff}$, $\Delta F_{aci}^{eff}$, $f_c$, or $L$. We express the sensitivities in relative form to account for the different mean values between the

simulations of Grosvenor and Carslaw (2020) and the coupled simulations. We find that the aerosol increase in Grosvenor and Carslaw (2020) causes a 59% increase in $N_d$, a 1.1% increase in $f_c$, a 0.82% increase in $L$ and a 2.7% increase in $F_{SW\uparrow}$. The $F_{SW\uparrow}$ increase was further split into a 2.2% increase from $\Delta F_{aci}^{eff}$ and a 0.5% increase from $\Delta F_{ari}^{eff}$ again based on the values of these from the nudged runs.

We can then use these percentage changes in the various quantities (denoted $\Delta x_{nudged}^{eff}(\%)$) and the percentage changes in

$N_d$ ($\Delta N_{d\,nudged}^{eff}(\%)$) to estimate the percentage changes in $x$ that would occur in the coupled simulations if the sensitivity of these quantities to $N_d$ were the same as in the Grosvenor and Carslaw (2020) simulations :-

$$\Delta x_{coupled}^{eff}(\%) = \frac{\Delta x_{nudged}^{eff}(\%)}{\Delta N_{d\,nudged}^{eff}(\%)} \Delta N_{d\,coupled}(\%)$$

(C1)

, where

$$\Delta N_{\text{d coupled}}(\%) = \frac{100 \Delta N_{\text{d coupled}}}{\overline{N_{\text{d coupled}}}} \tag{C2}$$

Here $\Delta N_{\text{d coupled}}$ is the change in $N_{\text{d}}$ over the period in question from the coupled run and $\overline{N_{\text{d coupled}}}$ is a mean value of $N_{\text{d}}$ from the coupled run used as a baseline for the relative changes. Here we use the mean over first 5 years of the period. $\Delta x_{\text{coupled}}$ can then be calculated using

$$\Delta x_{\text{coupled}} = \frac{\Delta x_{\text{coupled}}(\%) \overline{x_{\text{coupled}}}}{100} \tag{C3}$$

where $\Delta x_{\text{coupled}}(\%)$ comes from Eqn. C1 and $\overline{x_{\text{coupled}}}$ is the mean value of $x$ from the coupled run taken over the first 5
years of the period.

## Appendix D:  Summation of terms in Feedback figures

The difference between the overall change in a quantity X in the AerChemMIP-aerosol-only-proxy experiment is denoted as $\Delta X_{\text{aer}}$ and is estimated as the difference between the all-emissions experiment (AerChemMIP-all-emissions) and the GHG-only proxy (AerChemMIP-GHG-only-proxy, see Section 2.2.2) :-

$$\Delta X_{\text{aer}} = \Delta X_{\text{all}} - \Delta X_{\text{GHG}} \tag{D1}$$

For the AerChemMIP and DAMIP aerosol-only experiments feedbacks are estimated from the difference between $\Delta X_{\text{aer}}$ and the change due to aerosol ERF ($\Delta X_{\text{aer}}^{\text{eff}}$) following Eqn. 2. For the AerChemMIP experiments, following Eqn. D1, Eqn. 2 becomes :-

$$\Delta X_{\text{feedback aer}} = \Delta X_{\text{aer}} - \Delta X_{\text{aer}}^{\text{eff}} = \Delta X_{\text{all}} - \Delta X_{\text{GHG}} - \Delta X_{\text{aer}}^{\text{eff}} \tag{D2}$$

The total feedback is calculated in Figs. 10, 11 and 12 as $\Delta X_{\text{feedback aer}}$ plus the feedback from the GHG-only experiment ($\Delta X_{\text{feedback GHG}}$). Since the feedback for the GHG-only experiment is assumed to be $\Delta X_{\text{GHG}}$ we have (using Eqn. D2) :-

$$\Delta X_{\text{feedback tot}} = \Delta X_{\text{feedback aer}} + \Delta X_{\text{feedback GHG}} = \Delta X_{\text{feedback aer}} + \Delta X_{\text{GHG}} = \Delta X_{\text{all}} - \Delta X_{\text{aer}}^{\text{eff}} \tag{D3}$$

It then follows that the total feedback plus the aerosol forcing is :-

$$\Delta X_{\text{feedback tot}} + \Delta X_{\text{aer}}^{\text{eff}} = \Delta X_{\text{all}} \tag{D4}$$

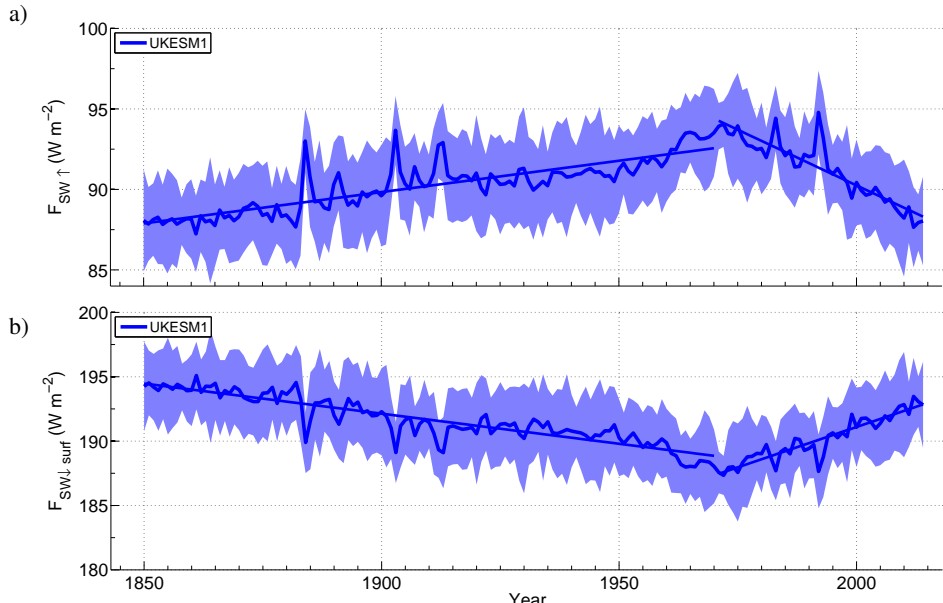

**Figure E1.** Timeseries of $F_{SW\uparrow}$ for the UKESM1 as in Fig 2 (top), but also showing the surface downwelling SW flux timeseries (bottom).

, such that this term is exactly equal to the change in X from the all-emissions run. However, this does not apply to the DAMIP results for which Eqn. D1 does not apply.

## Appendix E: Implications for surface SW downwelling fluxes

Here we consider the implications of the results analysing the SW flux at TOA ($F_{SW\uparrow}$) for downwelling surface SW fluxes ($F_{SW\downarrow surf}$). $F_{SW\downarrow surf}$ is important for a variety of reasons, for example, it more directly relates to the energy input into the
surface, which may affect ocean heat storage, surface temperatures, the AMOC, etc. From Fig. E1 it is clear that trends in $F_{SW\downarrow surf}$ follows a very similar pattern to those of $F_{SW\uparrow}$ except in a mirror image. This hints that the results for $F_{SW\uparrow}$ described in this paper are likely to be applicable to $F_{SW\downarrow surf}$. However, further analysis would be needed to definitively prove this.

*Data availability.* All data used is publically available online.

*Author contributions.* DPG analysed the model data and produced the text and figures. KSC helped to analyse the model output and provided feedback and edits to manuscript drafts.

*Competing interests.* The authors declare that they have no conflict of interest.

*Acknowledgements.* DPG was supported by the National Environmental Research Council (NERC) national capability grant for The North Atlantic Climate System Integrated Study (ACSIS) program (grant NE/N018001/1) via NCAS. We acknowledge use of the MONSooN system, a collaborative facility supplied under the Joint Weather and Climate Research Programme, a strategic partnership between the Met Office and the Natural Environment Research Council.

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
