# Peer review of "Change from aerosol-driven to cloud feedback-driven trend in shortwave radiative flux over the North Atlantic"

_Atmospheric Chemistry and Physics, 2022_

## Author Comment (AC1)

We would like to thank both reviewers for taking the time to read our paper and providing constructive and very useful feedback.

**Reviewer #1**

In this study, the authors used the HadGEM3/UKESM1 climate models to understand how and why shortwave fluxes have changed over the North Atlantic Ocean over the CMIP6 historical period (1850-2014). They identify two periods where the trend in outgoing shortwave flux, and the causes for that trend, differ. The first period is 1850-1970, characterized by an increase in outgoing shortwave flux, which the authors link in the model to an aerosol-driven increase in cloud droplet number. The second period, 1970-2014, sees a decrease in outgoing shortwave flux, explained by a feedback of greenhouse gas warming on cloud fraction. The analysis also contains a comparison of the models to relevant observations, and comparisons between different simulations.

The analysis is very thorough and proceed in well-defined steps. Figures illustrate the discussion well, and Table give detailed numbers. The paper is very well written, although is a challenging read because of the high density of information that is discussed. The findings have interesting implications on the use of observations to constrain aerosol forcing, given that concurrent, non-aerosol cloud feedbacks are also present.

Given that HadGEM3 and UKESM1 are tightly related, the study is not far from being a single-model study, but that is justified because the depth of analysis and the need for additional simulations make the work difficult to replicate in a multi-model context. The methodology is interesting, with a complicated gymnastic of double differences between simulations, especially in section 3.5.

The main weakness of the study, which is implicitly acknowledged by the authors, is that the implications for the real world are difficult to identify. The comparison to observations arrives late and potentially invalidates a lot of what the manuscript said up to that point. We know that observations of trends are unfortunately insufficient to constrain aerosol radiative forcing and climate sensitivity, so the authors cannot satisfactorily unravel (lines 619-623) what the comparison to observations means for the preceding analysis. Similarly, the differences between the AerChemMIP and DAMIP simulations discussed in Appendix A, which are not really understood, make the findings fragile.

I am not sure how to mitigate that weakness. The paper could be built the other way around, perhaps, dealing with model-observations and model-model differences first. That could force the discussion to account for the implications more explicitly. But it will always be the case that results and discussions in sections 3.5 and 4 leave many questions open. Perhaps simply be more upfront in acknowledging the issue in the abstract and conclusion?

We agree that it is difficult to make firm conclusions about what the model vs observations tells us about model weaknesses (i.e., too-strong feedbacks, too-strong aerosol forcing, or something else). However, we still believe that the results give us some degree of insight and that the conclusions of the paper hold, given some caveats. We discussed many of the caveats mentioned by the reviewer already in the paper; e.g., lines 619-623, where we discussed the impact on the conclusions of the model feedback potentially being too strong, and lines 575-579 where we acknowledged the uncertainties introduced by these issues and by the pattern effect :-

$\Delta F_{SW}$ (converted to a trend for Fig. 13) of the all-emissions runs (UKESM1 and HadGEM). This hints that the magnitude of the SST change may be more important than the spatial pattern for $\Delta F_{SW}$ in the N. Atlantic leaving open the possibility that the cloud feedbacks or aerosol forcing in the model are incorrect. However, the uncertainties are large and further work is needed to determine this.

However, we have now provided extra quantification to back up our claim that the conclusion of feedbacks dominating over aerosol forcing in the post-1970 period is still likely to be valid despite the possibility that the model feedbacks are too strong; the ΔSW from feedbacks would have to reduce in magnitude from -5.4 to -0.83 W/m2 in order for the feedbacks to be of the same strength as the forcing over that post-1970 period.  Here is the revised text :-

[revised manuscript text omitted]

Other comments:
Line 166: "as realistic as possible" is an optimistic view of the process of creating historical emission datasets. It should be noted that CEDS emissions have gone through many revisions, which show sizeable changes even in sulfur dioxide emissions compared to the version documented in Hoesly et al (2018). The simulations presented here may not use the latest version.

The text has been changed as follows :-

runs that were performed for CMIP6 (Sellar et al., 2019; Williams et al., 2017; Kuhlbrodt et al., 2018). These ran from 1850 to 2014 using greenhouse gas (GHG), aerosol, natural (e.g., volcanic) and other emissions that  were designed to represent the real emissions over this period. We note that there are likely to be uncertainties in these emissions that will lead to model errors. The ensembles were designed to capture a range of possible ocean and atmospheric states in order to sample the natural multi-decadal variability.

Line 228: Nit-pick, but cloud fraction does not affect cloud albedo. It affects planetary albedo.
This has been changed as follows :-

in trace gas concentrations. However, we do not attempt to separate these effects here. For changes in  the all-sky (i.e., combined cloudy and clear regions) albedo, we consider the effect of changes in the three main variables that affect it, namely cloud fraction ($f_c$), cloud droplet number concentration ($N_d$) and in-cloud liquid water path ($L$), along with $F_{SW\uparrow}$ from the clear-sky regions above clouds and also $A_{surf}$ in cloudy-sky conditions. However,

Table 1: Which are the HadGEM simulations?
Sorry, this is a mistake in the caption that has been rectified – there are no HADGEM values in Table 1.

Captions of Figures 1 and 3: Why "intermodel standard deviation"? There is only one model in that calculation.
The word "intermodel" has been removed from these captions.

Lines 301-304 and Table 2: Any idea why the offline calculations do a poorer job after 1970? Offline calculations must miss an input that becomes important then. Perhaps something to do with water vapour or gaseous absorption? And offline calculations seem to have less variability – I suppose that is to be expected?
It is difficult to assess what might be causing the poorer performance of the offline calculations after 1970 relative to pre-1970. The offline calculations use the online SWTOA clear-sky fluxes for the cloud-free regions rather than attempting to calculate the clear-sky fluxes from the input variables. Therefore the calculations might be expected to include the influence of water vapour or gaseous absorption. However, it is possible that there are some interactions between those processes and clouds that is not captured. There is a small estimated (from the offline calculations) negative contribution (-0.5 W/m2) to the outgoing TOA SW flux due to changes in the clear sky outgoing SW TOA flux in the all-emissions run that might suggest some influence from water vapour or gaseous absorption. Although we note that the clear-sky contribution is smaller in the GHG-only proxy run (-0.2 W/m2) when these effects might be expected to be larger. Other possible explanations are that there are larger changes in other factors during the post-1970 period, such as high altitude/ice cloud amount.
        However, we also note that the overall deltas for the online and offline calculations of SW agree within the uncertainties for the post-1970 period and that the uncertainties are also larger for this period.
        It is probably to be expected that the variability of the offline calculations is lower since the calculations are using monthly averages rather than data at the model timesteps as used by the online calculations.

Line 380: The lack of an impact from natural aerosols is partly by construction. Sea salt is the natural aerosol most likely to affect liquid cloud albedo. Its historical trend in the North Atlantic region is probably limited to locations where sea ice becomes open oceans, but those regions are excluded from the analysis.
Yes, this may be true and a similar point was brought up by Reviewer#2 regarding climate feedbacks for sea-salt aerosols. There are only small trends in droplet concentrations ($N_d$) in the GHG-only proxy run (Table 1) for which there are large temperature changes (and no anthropogenic aerosol emissions); $\Delta N_d$ = -1.0±0.49 cm$^{-3}$ for the pre-1970 period and 0.54±0.91 cm$^{-3}$ for the post-1970 period. We also computed $\Delta N_d$ values in just the sea-ice region for the GHG-only run; these were 0.57±0.47 cm$^{-3}$ and -0.84±0.74 cm$^{-3}$ for the pre- and post-1970 periods, respectively. Hence, there is likely to be only a small sea-salt aerosol trend effect even in the sea-ice regions, at least according to the model.

Although we note that line 380 is referring to the natural aerosol-only DAMIP experiment and so refers to the forcing effect of the included natural aerosols, which is mostly volcanic aerosol emissions.

We have changed the text to discuss these issues as follows :-

In this section we perform the same analysis as in Section 3.2 to quantify how much the individual changes in aerosol and cloud properties contribute to $\Delta F_{SW\uparrow}$ except for the single-emissions experiments (aerosol-only and GHG-only). It is clear from the DAMIP experiment results Figs. A1 and A3, and Table A1 (see Appendix A) that  the DAMIP natural aerosol forcing, which comes mostly from volcanic aerosols, has almost no influence on the $F_{SW\uparrow}$ trends, therefore we do not consider them further. However, there could be influences from natural aerosols that are not captured by the DAMIP natural emissions such as feedbacks between sea-spray CCN and temperature. Some of these will be represented in the experiments used here such as the effects on sea-spray from changes in wind speed as a result of temperature change. Our results (Table 1 and Fig. 5d) show that there is little change in $N_d$ in the AerChemMIP-GHG-only-proxy experiment (-1.0±0.49 $cm^{-3}$ for the pre-1970 period and 0.54±0.91 $cm^{-3}$ for the post-1970 period) in which large temperature changes occur, which suggests little influence of climate feedbacks on CCN. Our results are likely to exclude the impact of changes in sea-spray due to changes in sea-ice coverage since we deliberately excluded sea-ice covered regions. Therefore we calculated the changes in $N_d$ for only the sea-ice regions and found values of 0.57±0.47 and -0.84±0.74$cm^{-3}$ for the pre- and post-1970 periods, respectively, suggesting that the effect is small for this model.

Lines 409-413: The definition of climate feedbacks introduced here is unclear. Does that include rapid adjustments? The remainder of the work suggests adjustments are excluded, so I suggest rephrasing to make that clearer. IPCC practice is to see feedbacks as the climate response mediated by a large-scale change in temperature – that is, excluding rapid adjustments by definition.

The rapid adjustments to aerosols is excluded from the feedback term since they are included in the aerosol forcing term that is subtracted. However, rapid adjustments to CO2 are included in the feedback term, but are likely to be small (as also discussed in the next point). The text in this paragraph has been altered to make this clearer and to address the next point raised by the reviewer :-

**3.5  Aerosol forcing vs cloud-climate feedbacks**

Here we examine the relative roles of aerosol forcing and feedbacks resulting from climate change (temperature, atmo-
430  spheric/ocean circulation changes, etc.) on the change in $F_{SW\uparrow}$ and the cloud variables.

Aerosol forcing  is the change in $F_{SW\uparrow}$ caused by a change in aerosols without a change in
climate (SSTs; water vapour; atmosphere and ocean circulation, etc.). ~~The remainder of the change in  is then assumed to be
caused by changes in climate (termed climate feedbacks here). Changes in~~ This includes rapid cloud adjustments of $f_c$ and $L$
 which are potentially a major cause of changes in $F_{SW\uparrow}$,
435  ~~(without a change in temperature) as well as cloud feedbacks. Here, we aim to attribute the changes in  and $L$ to these two
causes using a similar method to that just described for .~~

For the greenhouse gas-only runs, we assume that the changes in $F_{SW\uparrow}$, $f_c$ and $L$ are due to climate feedbacks with no effect
of  greenhouse gases on cloud or clear-sky adjustments. However, we ac-
knowledge that such  effects may be possible
440  . For example, the results of Andrews and Forster (2008) showed a -0.18 W m$^{-2}$ global change in
$F_{SW\uparrow}$ from greenhouse gas adjustments (termed semi-direct forcing in that paper) for the HADGEM1 model in a doubling $CO_2$
experiment. This would represent a small fraction (6.4%) of the -2.8 W m$^{-2}$ change from the AerChemMIP-GHG-only-proxy
run for the post-1970 period (although the latter is for the North Atlantic region only) and is also likely to be an overestimate
for our case since the change in $CO_2$ for the post-1970 period is less than a doubling. Furthermore, Figure 7.4 of the AR6

445  assessment (Forster et al., 2021) estimates the global $CO_2$ adjustment effect to be around 5% of the total ERF, although this is
for the combined shortwave and longwave values.

Line 416: I suggest updating that discussion by basing it on the AR6 assessment, for example
Table 7.4 on CO2 rapid adjustments. That will not change the conclusion that rapid adjustments
to aerosol forcing are larger than those to greenhouse gases.
Thanks for pointing us towards this. We have added in some discussion based on the AR6 report
(Figure 7.4), although there is no separation between SW and LW effects there, which makes
interpretation more difficult. We have also added in more detail from the Andrews and Forster
(2008) paper (see the revised text above for the last point).

Caption of Figure 9: I do not think that this caption is the place for explaining how feedbacks are
estimated. I would suggest adding a diagram or a table showing how forcing, feedbacks etc. are
separated from double differences between pairs of simulations.
We have created a schematic figure to show how the feedbacks, etc. are calculated:-

[Figure]

**Figure 1.** Schematic showing how various quantities are calculated. Some of the quantities are not introduced until later in the paper. The same methodology also applies to the DAMIP (HadGEM-based) results except that GHG-only and aerosol-only proxies are not required (Box 1) since there are dedicated experiments with GHG-only and aerosol-only emissions.

We have also added an equation for the "Aerosol-feedback" term and now refer to this in the caption for Fig. 9 (now Fig. 10). The methods detail in the caption has been removed and added to the text where necessary. The caption now reads :-

[Figure]

**Figure 10.** The relative roles of aerosol forcing and climate feedbacks in explaining $\Delta F_{SW\uparrow}$ between 1870 and 1970 (left column) and between 1971 and 2014 (right) for the AerChemMIP UKESM1 runs. "Aerosol-only" is the change in the AerChemMIP-aerosol-only-proxy runs as in Fig. 7 ($\Delta F_{SW\uparrow \, aer}$). "ACI" and "ARI" are the aerosol effective radiative forcings ($\Delta F^{eff}_{ari \, coupled}$ and $\Delta F^{eff}_{aci \, coupled}$). "Aerosol Feedback" is the climate feedback term for the AerChemMIP UKESM1 runs ($\Delta F_{SW\uparrow \, feedback \, aer}$) calculated using Eqn. 2 and "Aerosol Feedback from $\Delta T$" ($\Delta F^{\Delta T}_{SW\uparrow \, feedback \, aer}$) is that calculated using Eqn. 3. "Total (Aerosol+GHG) Feedback" is the estimated total climate feedback in the all-emissions run ($\Delta F_{SW\uparrow \, feedback \, tot}$) calculated by summing $\Delta F_{SW\uparrow}$ from the AerChemMIP-GHG-only-proxy run ($\Delta F_{SW\uparrow \, feedback \, GHG}$) and $\Delta F_{SW\uparrow \, feedback \, aer}$. Also shown is $\Delta F_{SW\uparrow}$ for the all-emissions UKESM1 AerChemMIP runs (AerChemMIP-all-emissions). Arrows on the left plot are drawn to indicate values that add together to give other values on the plot (see Eqn. 2 and Appendix D). These also apply to the righthand panel and to all panels for Figs. 11 and 12, but are omitted for clarity. Arrows for $\Delta F^{eff}_{aer \, coupled} = \Delta F^{eff}_{ari \, coupled} + \Delta F^{eff}_{aci \, coupled}$ have also been omitted.

We have also added arrows to indicate the quantities that add together to make other quantities as requested by Reviewer #2.

Line 534: Could cite Andrews et al. (2019)
https://agupubs.onlinelibrary.wiley.com/doi/full/10.1029/2019MS001866 here. Mid-latitude cloud feedbacks are indeed listed as a possible source of the large ECS in HadGEM3/UKESM1.
Thanks, this has now been added.

Line 578: It should be noted here that the strength aerosol forcing has been decreased during model development (Mulcahy et al. 2019). It does not follow that aerosol forcing is still too strong, but it shows that an excessive strength is a long-standing concern of the developers of the model.
Thanks, we have added this to Conclusions+Discussion section at the end of this paragraph (citing Mulcahy et al. 2018, which seems the most appropriate reference) :-

feedback and aerosol forcing effects to be equal. Therefore, the conclusion is likely to remain robust. On the other hand, if the model aerosol forcing is too large then using the correct aerosol forcing would enhance the ratio between cloud feedback and aerosol forcing and hence strengthen the conclusions. Furthermore, the strength of the aerosol forcing was decreased during UKESM1 model development (Mulcahy et al., 2018) showing that an excessive forcing strength is a long-standing concern of the model developers.

Technical comments:
Line 135: Typo cooilng
Line 234: I do not think that tau_c and r_e have been defined, but I may have missed them.
Line 249: Parenthesis is not closed.
Caption of Figure 13: Typo atmosphere
Line 665: Typo simlar
Line 712: Missing words "lack reduction"
Line 782: Typo tempeatures

All of the above have been fixed, thanks for spotting them.

**Reviewer #2**

The authors present a very thorough investigation into the behaviour of shortwave radiation fluxes above the North Atlantic, over the historical era, in two versions of the UK climate model (UKESM, HadGEM3). They find two regimes with markedly different behaviour; before and after 1970; and attribute them (primarily) to an increase in aerosol concentrations and cloud responses to surface temperature change, respectively.

Overall, this is an impressively detailed study, with well described reasoning and broad ranging but established methods. It reads almost like a textbook at times, taking the reader through all main factors thought to be able to influence F_SW and disentangling their various influences. The analysis and the manuscript are clearly very well worked through, and I therefore have very little to offer in terms of deeper feedback. This paper could well be published as-is, and should certainly not require more than a minimal revision.

Some minor questions and comments:

   * If I have one concern with the paper, it is that it is very long and at times quite wordy. There is a risk that the nice and highly instructive results get lost because the community doesn't have

time to read through it all. Therefore: Would it be an idea to include a process level schematic of the factors contributing before and after 1970? I.e. an annotated version of Figure 1a, with some arrows and icons, to show what is changing and why?

Thanks for the idea – we have added a schematic as you suggested as the final figure of the paper :-

[Figure]

**Figure 15.** Schematic showing the main influences on the determination of the change in $F_{SW\uparrow}$ during the pre-1970 and post-1970 periods. The quoted percentages are the percentage contributions to $\Delta F_{SW\uparrow}$ from aerosol forcing and climate feedbacks for the two periods.

* Unless I'm missing it, I don't think you discuss the temperature feedback on sea salt aerosols as a potential contributor to F_SW trends? This effect should be there for the North Atlantic, at least for the post-1970 period, somewhat counteracting the reduction in anthropogenic CCN. (You have the DAMIP natural forcer experiments, and show that it can be ignored in this context, but that simulation will not have the natural aerosol feedbacks.)

As noted in the response to Reviewer#1, there is little trend in droplet concentrations in the GHG-only proxy run for which there are large temperature changes (and no anthropogenic aerosol emissions) suggesting little impact from temperature feedbacks on CCN, at least according to the model.

* In section 2.3, I can't quite see that you've quantified the impact of excluding grid boxes with sea ice formation. Presumably the effect is small, but could it introduce some biases? (Domination of southern grid boxes, or spurious seasonality?)

We removed sea-ice regions because sea-ice is highly variable between simulations and we were concerned that when it is present the results may be more influenced by this variability rather than by the factors of interest, e.g., aerosol forcing and climate feedbacks.  However, we have tested the impact of including the sea-ice regions on the shortwave flux trends. For the pre-1970 period it changes the ΔF_SW from 4.7 to 4.8 W/m2 and for the post-1970 it changes it from -6.0 W/m2 to -6.1 W/m2 for the UKESM1 ensemble mean. Therefore, the impact is likely to be negligible.

It's unlikely that there would be any spurious seasonality since the sea-ice grid boxes were removed for the whole timeseries (and the same grid-boxes were removed for all the models).

* Figure 1, and others: Would it be worth also showing NA AOD? Simply because there are so many other studies that use AOD, and therefore it becomes easier to compare your results to theirs?
 We have now added AOD to Figures 1, 5, 6, 12, 13, A1 and A3 and in the appropriate tables. The results are consistent with those of $N_d$.

* Figures 4, 7, 8, ...: In many of these, one dot is the net of others. Could this be highlighted more clearly, with colors, symbols or similar? (You do this for aerosols in places, but I still struggled a bit to understand how all the factors summed up - or not - in the various figures.)
We have now added arrows to the first panel of Fig. 9 (now Fig. 10) to show this, along with the addition of equations in the text to make this clear.

[Figure]

**Figure 10.** The relative roles of aerosol forcing and climate feedbacks in explaining $\Delta F_{SW\uparrow}$ between 1870 and 1970 (left column) and between 1971 and 2014 (right column) for the AerChemMIP UKESM1 runs. "Aerosol-only" is the change in the AerChemMIP-aerosol-only-proxy runs as in Fig. 7 ($\Delta F_{SW\uparrow\,aer}$). "ACI" and "ARI" are the aerosol effective radiative forcings ($\Delta F^{eff}_{ari\ coupled}$ and $\Delta F^{eff}_{aci\ coupled}$). "Aerosol Feedback" is the climate feedback term for the AerChemMIP UKESM1 runs ($\Delta F_{SW\uparrow\,feedback\,aer}$) calculated using Eqn. 2 and "Aerosol Feedback from $\Delta T$" ($\Delta F^{\Delta T}_{SW\uparrow\,feedback\,aer}$) is that calculated using Eqn. 3. "Total (Aerosol+GHG) Feedback" is the estimated total climate feedback in the all-emissions run ($\Delta F_{SW\uparrow\,feedback\,tot}$) calculated by summing $\Delta F_{SW\uparrow}$ from the AerChemMIP-GHG-only-proxy run ($\Delta F_{SW\uparrow\,feedback\,GHG}$) and $\Delta F_{SW\uparrow\,feedback\,aer}$. Also shown is $\Delta F_{SW\uparrow}$ for the all-emissions UKESM1 AerChemMIP runs (AerChemMIP-all-emissions). Arrows on the left plot are drawn to indicate values that add together to give other values on the plot (see Eqn. 2 and Appendix D). These also apply to the righthand panel and to all panels for Figs. 11 and 12, but are omitted for clarity. Arrows for $\Delta F^{eff}_{aer\ coupled} = \Delta F^{eff}_{ari\ coupled} + \Delta F^{eff}_{aci\ coupled}$ have also been omitted.

* Your convention is that F_SW is upwelling; I got this after reading a bit, but I don't think you explicitly define it? Maybe have it already in the abstract, line 5? ("positive upwelling F_SW trend"?)
We have added the word "upwelling" to the definition in the abstract and have also changed the symbol for upwelling F_SW to include an up arrow to make this clear throughout.

* The last reference (Zhou et al. 2016) comes twice in the ref list.
Thanks, this has been removed.

Thanks for a very interesting paper.
You're welcome, glad that you found it interesting and thanks for your comments!